# The Polygenic Risk Score Knowledge Base offers a centralized online repository for calculating and contextualizing polygenic risk scores

Madeline L. Page [1,84], Elizabeth L. Vance[1,84], Matthew E. Cloward [2,84], Ed Ringger[2], Louisa Dayton[2], Mark T. W. Ebbert[1,3,4], the Alzheimer's Disease Neuroimaging Initiative*, Justin B. Miller [1,3,5,85] & John S. K. Kauwe [2,85 ✉]

The process of identifying suitable genome-wide association (GWA) studies and formatting the data to calculate multiple polygenic risk scores on a single genome can be laborious. Here, we present a centralized polygenic risk score calculator currently containing over 250,000 genetic variant associations from the NHGRI-EBI GWAS Catalog for users to easily calculate sample-specific polygenic risk scores with comparable results to other available tools. Polygenic risk scores are calculated either online through the Polygenic Risk Score Knowledge Base (PRSKB; https://prs.byu.edu) or via a command-line interface. We report study-specific polygenic risk scores across the UK Biobank, 1000 Genomes, and the Alzheimer's Disease Neuroimaging Initiative (ADNI), contextualize computed scores, and identify potentially confounding genetic risk factors in ADNI. We introduce a streamlined analysis tool and web interface to calculate and contextualize polygenic risk scores across various studies, which we anticipate will facilitate a wider adaptation of polygenic risk scores in future disease research.

[1] Sanders-Brown Center on Aging, University of Kentucky, Lexington, KY, USA. [2] Department of Biology, Brigham Young University, Provo, UT, USA. [3] Division of Biomedical Informatics, Department of Internal Medicine, University of Kentucky, Lexington, KY, USA. [4] Department of Neuroscience, University of Kentucky, Lexington, KY, USA. [5] Department of Pathology and Laboratory Medicine, University of Kentucky, Lexington, KY, USA. [84]These authors contributed equally: Madeline L. Page, Elizabeth L. Vance, Matthew E. Cloward. [85]These authors jointly supervised this work: Justin B. Miller, John S.K. Kauwe. *A list of authors and their affiliations appears at the end of the paper. ✉email: kauwe@byu.edu

Genome-wide association (GWA) studies have revolutionized the study of complex diseases and trait heritability by identifying genome-wide significant genetic loci associated with specific phenotypes. Tens of thousands of genetic associations are currently implicated in diseases or traits with genome-wide significance (p-value $<5 \times 10^{-8}$)[1], and additional associations have been discovered through meta-analyses[2–4]. These GWA studies span various complex diseases and traits[5–7], including major depressive disorder[8], type 2 diabetes mellitus[9], Alzheimer's disease[10], coronary artery disease[11], schizophrenia[12], numerous cancers[13–15], lifestyle choices (e.g., smoking, drinking, etc.[16,17]), and have helped identify candidate drug targets[18–20].

GWA studies are effective at identifying individual genetic locus-trait associations. However, GWA results on their own cannot determine the total genetic liability for a given trait in a genome of interest. Polygenic risk scores utilize GWA summary statistics to quantify the aggregate genetic risk for a disease or trait based on all associated genetic variants present in a genome[21].

Accordingly, polygenic risk scores are dependent on the underlying summary statistics from a GWA study. However, most large-scale GWA studies have been conducted on predominantly European populations[22], with results that often do not translate to other populations[23] due to differences in allele frequencies and linkage disequilibrium patterns[24–26]. For instance, effect sizes reported in GWA studies performed primarily on populations of European descent were found to be significantly higher than corresponding effect sizes reported by GWA studies consisting entirely of non-European individuals[27]. The lack of diversity in GWA study cohorts can also cause important risk alleles in minority populations to remain unidentified. For example, the Population Architecture using Genomics and Epidemiology (PAGE) study found that a novel risk variant associated with the number of cigarettes smoked per day existed at a frequency of 17.2% in Native Hawaiian participants but was absent or rare in most other populations[28].

Choosing an appropriate GWA study to calculate polygenic risk scores is paramount to the fidelity of the calculations because the accuracy and predictive power of a polygenic risk score is dependent on the power and scope of the corresponding GWA study data[29,30]. When used appropriately, polygenic risk scores can capture genetic predisposition for diseases or traits across various genetic markers and can be used to assess the genetic risk compared to a specific population[31–34]. Because polygenic risk scores can stratify populations based on distinct risk, they can be useful in determining clinical and personal interventions[35,36]. For example, a polygenic risk score can greatly inform cancer risk management for *BRCA1* carriers, who have a 21% risk of developing breast cancer by age 50 if they are in the lowest polygenic risk score decile for breast cancer and a 39% risk of developing breast cancer by age 50 if they are in the highest polygenic risk score decile[37]. Likewise, polygenic risk scores can be used to classify disease subtypes[36,38,39], and differences in polygenic risk scores for epilepsy reliably correspond to the variation in epilepsy subclassifications[40,41]. Furthermore, polygenic risk scores can effectively explore genetic overlap between pairs of traits[42], which has revealed a shared genetic basis for multiple pairs of psychiatric disorders[43,44]. Surprisingly, polygenic risk scores are also able to show a lack of correlation in pairs of neurological traits, such as multiple sclerosis and amyotrophic lateral sclerosis, where genetic correlation might otherwise be expected[45]. Polygenic risk scores can also test for gene-by-environment and gene-by-gene interactions[46,47] through Mendelian randomization studies, which detect causal genetic relationships[48,49], and genotype-by-environment interactions based on GWA summary statistics are increasingly common on biobank-scale data[50].

There currently exists a spectrum of tools available for calculating polygenic risk scores, ranging from direct-to-consumer genetics companies (e.g., 23andMe[51]) to downloadable software packages (e.g., PRSice-2[52]). PRSice-2 is a multi-faceted tool that greatly facilitates polygenic risk score analyses of large cohorts compared to alternative software such as LDpred[53] and lassosum[54]. However, PRSice-2 requires users to have an in-depth knowledge of bioinformatics, supply their own GWA summary statistics, use bgen or binary PLINK[55] file formats for genetic data (i.e., no VCF files), and perform all calculations locally (i.e., no dedicated server for testing and/or small datasets). Further, PRSice-2 requires all variants to be annotated with the same accession numbers as the GWA study, so merged or deprecated accession numbers are not identified using PRSice-2. PRSice-2 also has a significant learning curve to understand and utilize the available options, which can limit its application in labs without a strong bioinformatics presence. These constraints have potentially limited the application of polygenic risk score calculations in assessing off-target disease susceptibility and the wider adaptation of polygenic risk scores in other genetic analyses.

Other notable efforts to centralize polygenic risk scores for research, such as the Polygenic Score Catalog (PGS Catalog)[56] and Impute.me[57], have greatly improved the interpretability and dissemination of polygenic risk scores on precomputed data. However, they currently lack the capability of performing high-throughput analyses on user-specific data across all available studies. Additionally, users are required to select specific studies or traits to analyze a priori, which makes data exploration much more time consuming.

Here, we present the Polygenic Risk Score Knowledge Base (PRSKB), a web server (https://prs.byu.edu) and command-line interface for calculating polygenic risk scores using various GWA summary statistics and a single command at runtime. As of March 16, 2022, the PRSKB contains the following data that can be used for user-specific calculations of polygenic risk scores and contextualization against larger cohorts: 250,134 variant associations; 125,433 unique single nucleotide polymorphisms; 20,798 unique study and trait combinations; 10,366 GWA study identifiers; and 3463 PubMed identifiers. We use genomic datasets from the 1000 Genomes Project[58], UK Biobank[59], and the Alzheimer's Disease Neuroimaging Initiative (adni.loni.usc.edu) to create polygenic risk score percentiles against which individual risk scores can be examined. We show that the PRSKB performs similarly to PRSice-2 and can accurately differentiate between Alzheimer's disease cases and controls in the ADNI dataset. Because the PRSKB simplifies polygenic risk score calculations and contextualization across thousands of studies that can all be performed with a single command at runtime, we anticipate that this tool will enable a wider adaptation of polygenic risk score calculations through clinical trial screenings, analyses of comorbidities, identifying confounding genetic factors, and various other analyses related to disease genetics.

## Results

We developed the PRSKB to simplify the process of calculating polygenic risk scores across all available GWA studies. Users can calculate polygenic risk scores through the user-friendly online calculator or command-line interface. The PRSKB GWA Study Browser allows users to identify which GWA studies can be used to compute polygenic risk scores and provides references for each study. Polygenic risk scores can be contextualized against the UK Biobank, population-specific 1000 Genomes data, and the ADNI dataset for each study in the database. The depth and breadth of studies in the database, as well as the collection of previously-

calculated risk scores from a variety of populations, facilitates the implementation of the PRSKB in future trait and disease research.

**Online polygenic risk score calculator**. The PRSKB calculator can calculate polygenic risk scores for multiple traits and studies. To run the calculator, users input target data either by typing reference RSID numbers and their corresponding alleles into a text box or by uploading a variant call format (VCF) file that stays on their browser and never reaches our database. Next, the user must specify the reference genome (hg38, hg19, hg18, or hg17) used to sequence the input variants if they are using the VCF file format so that the associations queried from the database correspond to the same reference assembly. By default, hg38 is used as a reference for RSIDs. Various filters allow users to choose specific studies, populations, or study types (e.g., users can choose to include only studies with the highest Altmetric score[60] or the largest study cohort reported by the GWAS Catalog, measured as the initial sample size plus the replication sample size). Finally, the user must designate a p-value threshold for GWA variants included in the calculations and whether they prefer a condensed or verbose output file. Supplementary Fig. 3 presents the PRSKB calculator interface.

The polygenic risk score results are written to a tab-separated values (TSV) output file presented in either a condensed or detailed format, or a JavaScript Object Notation (JSON) file (see Supplementary Fig. 4). Supplementary Data 4 and Supplementary Data 5 respectively show examples of the condensed and verbose output. Genetic variants with an odds ratio greater than one indicate an increased genetic risk of developing the disease or trait, while odds ratios less than one indicate genetic protection against the disease or trait. Similarly, beta values greater than zero increase genetic risk and beta values less than zero decrease genetic risk for the disease or trait.

Users can browse the GWA studies in our database to locate studies they wish to use in their calculations by searching for the first author, article title, trait, PubMed ID, or GWAS Catalog study accession ID. The GWA study browser can be accessed under the "Studies" tab on the PRSKB website or through "Option 2: Search for a specific study or trait" on the PRSKB CLI menu. Supplementary Fig. 5 introduces the GWA study browser interface. Alternatively, users can opt to use their own GWA study data, following the proper formatting requests listed on the PRSKB website or the PRSKB CLI menu.

**Command-line interface tool download**. In addition to the website, a downloadable command-line interface (CLI) tool is available for users to run the calculator directly from the command-line. This option is recommended for users running the calculator on multi-sample VCFs or calculating polygenic risk scores for more than 50 GWA studies. Required parameters include a path to the input file, a path to the output file, the p-value threshold for associations, the reference genome of the variants in the input file, and the superpopulation for the samples in the input file. Using only the required parameters, polygenic risk score calculations are run on every trait and study in the database. Optional parameters are used to filter which studies are included for calculations (e.g., specific traits, studies, or ethnicity of the study cohort). The CLI can also be run in two steps to perform large calculations without internet access, and it is multithreaded for improved computational efficiency (see Supplementary Fig. 6).

The CLI tool contains a built-in menu when run without parameters. This menu allows users to learn more about the CLI tool and the parameters required to run it, search the PRSKB database for traits and studies, view the usage statement, and run the risk score calculator (see Supplementary Fig. 7).

**The UK Biobank, 1000 genomes, and ADNI for polygenic risk score contextualization**. We present polygenic risk score distributions and summary statistics for each of the studies in the PRSKB database, generated from individual genetic data in the 1000 Genomes, UK Biobank, and ADNI datasets. Users can choose between the following cohorts as an approximate contextualization for their own reported risk scores: UK Biobank, 1000 Genomes—African, 1000 Genomes—American, 1000 Genomes—East Asian, 1000 Genomes—European, 1000 Genomes—South Asian, ADNI—Alzheimer's disease, ADNI—Mild Cognitive Impairment, and ADNI—cognitively normal. Polygenic risk score distributions on these precomputed data can be visualized as violin plots, box plots, or line plots of the percentile data. For example, Supplementary Fig. 8 depicts the distribution of polygenic risk scores for severe SARS-CoV-2 infection with respiratory failure for individuals in the UK Biobank cohort based on GWA summary statistics reported by Ellinghaus, et al.[61]. At this time, visualizations on the website are exclusively for precomputed scores and user-uploaded data are not graphed. However, percentile data can be found for user-uploaded data in the verbose output file.

**ADNI case study**. Although we used the GWA summary statistics from Jansen, et al.[2] to compare only two groups in the ADNI dataset due to limited sample size for the mild cognitive impairment group (i.e., we combined Alzheimer's disease or mild cognitive impairment versus controls and combined controls or mild cognitive impairment versus Alzheimer's disease), we used an adjusted significance level of 0.01 to account for multiple testing of five potential comparisons of Alzheimer's disease risk: Alzheimer's disease versus mild cognitive impairment; Alzheimer's disease versus controls; mild cognitive impairment versus controls; Alzheimer's disease or mild cognitive impairment versus controls; and mild cognitive impairment or controls versus Alzheimer's disease. A Mann-Whitney U test revealed a significant difference between Alzheimer's disease polygenic risk scores in individuals with a CDR ≥ 1 and individuals with a CDR ≤ 0.5 ($P = 2.75 \times 10^{-9}$). Similarly, a Mann-Whitney U test also detected a significant difference between Alzheimer's disease polygenic risk scores for individuals with a CDR = 0 and individuals with any amount of dementia (CDR ≥ 0.5), although it was less significant ($P = 1.97 \times 10^{-7}$). Figure 1 shows the comparisons of polygenic risk score distributions in each CDR cohort. Similar comparisons were made using GWA summary statistics from Lambert et al.[3] and Lo et al.[62], and are shown in Supplementary Figs. 9 and 10, respectively.

After calculating polygenic risk scores from all other studies in the PRSKB database for the individuals in the ADNI cohort and correcting for multiple testing, we identified 42 GWA studies that produced risk scores that significantly differ ($P < 4.21 \times 10^{-06}$) between individuals with and without Alzheimer's disease (see Supplementary Data 6) and found 29 GWA studies that produced risk scores that significantly differed ($P < 4.23 \times 10^{-6}$) between individuals with cognitive impairment and normal cognition (see Supplementary Data 7).

**Comparison to PRSice-2**. The PRSKB reports similar polygenic risk score results as PRSice-2. Figure 2a plots the polygenic risk scores calculated for both the PRSKB and PRSice-2 across ADNI participants using the Lambert, et al.[3] GWA study. Since polygenic risk scores are a relative measurement of genetic risk compared to a population, we compared the shape of the distributions from the

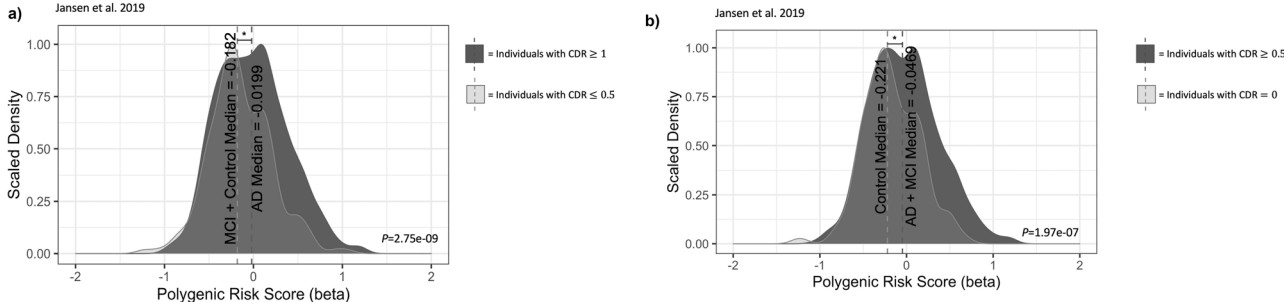

**Fig. 1 ADNI polygenic risk score distributions.** Alzheimer's disease polygenic risk score distributions are shown for **a** ADNI participants with a CDR ≥ 1 compared to ADNI participants with a CDR ≤ 0.5 and **b** ADNI participants with a CDR ≥ 0.5 compared to ADNI participants with a CDR = 0.

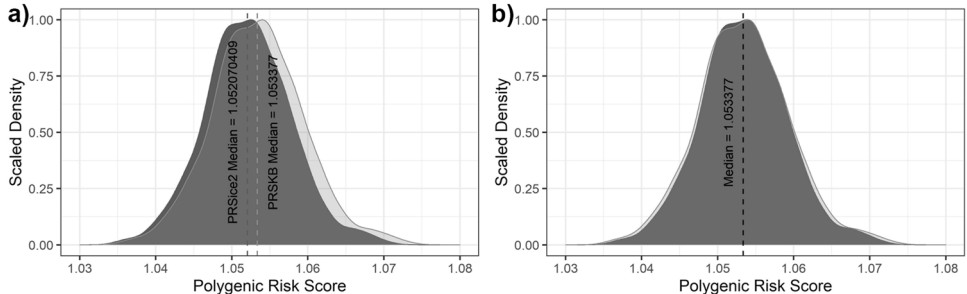

**Fig. 2 ADNI Polygenic Risk Scores using Lambert et al., 2013 GWA Summary Statistics.** PRSice-2 (dark grey), and the PRSKB (light grey) scores are shown. **a** PRSice-2 reports polygenic risk scores that center on 0, so 1.0 was added to each PRSice-2 score to put it on the same scale as the PRSKB, which centers polygenic risk scores based on odds ratios around 1.0. The PRSice-2 median score after transformation is 1.05207 and the PRSKB median score is 1.05338. **b** Since a polygenic risk score is a relative score compared to the sample population, we transformed the PRSKB scores by subtracting 0.00131 to overlap the shape of the distributions when both algorithms report the same median. Since the scores are normally distributed, a Welch's two-sample t-test was used to determine the similarity between the two distributions, which were nearly identical (t = 0.004782; P = 0.9962).

PRSKB and PRSice-2 to ensure that both algorithms report similar score distributions. After performing a minor transformation to have the same median values for both algorithms (original difference between medians is 0.001306), a Welch's two sample t-test shows that slight variations between the two algorithms do not change the overall shape of the distributions (see Fig. 2b; t = 0.004782; P = 0.9962). Similar comparisons between Alzheimer's disease and cognitive normal controls in the ADNI dataset using GWA studies from Lambert, et al.[3], Jansen, et al.[2], and Lo, et al.[62] show that the PRSKB and PRSice-2 produce very similar distributions (see Supplementary Figs. 11–13). Additionally, we found similar phenotypic variance explained by the PRSKB and PRSice-2 in ADNI when using associated variants in each of the three Alzheimer's disease genome-wide association studies (see Supplementary Table 5). The PRSKB was able to perform all polygenic risk score calculations using a single command at runtime, whereas PRSice-2 required individual input files for each study. Additionally, the PRSKB is a position-based tool and can handle mislabeled or merged accession numbers. This feature allowed the PRSKB to identify that variant *rs111418223* had been merged with *rs9271192* and labeled differently between ADNI and Lambert, et al.[3]. PRSice-2 was unable to automatically detect that those two variants had been merged because PRSice-2 depends on variant accession numbers. The PRSKB first searches for accession numbers, and then looks for chromosome and position pairs to identify associated variants in the target sequence.

## Discussion

The PRSKB is the bridge between GWA study data and calculating polygenic risk scores using user-specific datasets. Polygenic risk score calculations require GWA study summary statistics, yet current tools for calculating polygenic risk scores do not offer straightforward, comprehensive access to usable GWA study information. The PRSKB facilitates large-scale polygenic risk score analyses that currently (as of March 16, 2022) include 250,134 variant associations, 125,433 unique single nucleotide polymorphisms, 20,798 unique study and trait combinations, 10,366 GWA study identifiers, and 3,463 PubMed identifiers. These associations, which are automatically updated monthly from the GWAS Catalog, will likely enable researchers to identify previously unknown genetic biases in sampled cohorts and/or potential associations between traits.

The PRSKB improves polygenic risk score utilization by offering contextualization for individual risk scores. The UK Biobank, 1000 Genomes, and ADNI genetic risk score percentiles provide the information necessary for users to normalize their reported scores relative to large population-specific datasets.

The application of polygenic risk scores has become a critical resource in researching complex genetic diseases and personalized medicine. Although polygenic risk scores are effective at predicting genetic liability to a trait[31–34], risk prediction is not always the end objective to performing polygenic risk score calculations. Rather, these analyses are used for a wide variety of research purposes. Polygenic risk scores are useful at stratifying populations[35], influencing clinical and personal disease interventions[36,37], classifying disease subtypes[38,39], identifying genetic overlap between traits[42,44], and determining causal genetic relationships through Mendelian randomization studies[48,49,63]. Moreover, the implementation of polygenic risk scores has the potential to limit unknown covariates in future genetic studies by revealing individuals that have atypical genetic risk for phenotypes not directly studied.

Although polygenic risk scores have become increasingly prevalent in genetic research, historically, only minimal guidelines have existed for performing polygenic risk score analyses[21]. This limitation has led to inconsistencies in polygenic risk score methodologies in different studies and the misinterpretation of results. A recent publication by Choi, et al.[21] outlines a protocol for calculating polygenic risk scores, including detailed guidelines for performing and interpreting genetic risk score analyses. In our efforts to overcome the variability in current polygenic risk score research, we follow the standards set forth by Choi, et al.[21], including the implementation of the clumping and threshold (C + T) method. Furthermore, users are encouraged to follow the quality control measures for target and GWA data recommended by Choi, et al.[21] in order to ensure more optimal polygenic risk scores. Specifically, users are encouraged to ensure that the summary data and target samples are from the same population but avoid sample overlap or highly related samples. A target sample size of at least 100 and GWA study data with a SNP heritability ($h^2_{SNP}$) > 0.05 will also improve the power and accuracy of genetic risk score results[21]. Furthermore, we suggest that users who utilize the PRSKB to run bulk polygenic risk score analyses for post-hoc hypothesizing account for multiple testing when determining a significance threshold.

There are certain limitations to the PRSKB. For example, we remove multi-allele haplotype associations from the PRSKB database and ensure that combinations of multiple variants cannot have a single effect. The PRSKB analyzes each variant individually. Additionally, although LD clumping is the preferred method for the removal of variants in linkage disequilibrium[21], a common criticism of clumping is that the correlation and distance thresholds are generally arbitrarily chosen[21,64]. We selected threshold values that emulate clumping procedures performed in previous studies[64,65], but recognize that this choice may be an area for further development and research.

The PRSKB has other limitations that are inherent to GWA studies and polygenic risk score calculations[66]. A common limitation of GWA studies is their current inability to account for more than a small fraction of complex trait heritability[67]. Much of this missing heritability is attributed to rare variants or variants with small effect sizes that do not reach genome-wide statistical significance[68]. Incorporating rare variants in polygenic risk score calculations actually improves polygenic risk score prediction[69], and the PRSKB uses all associated variants in its calculations by default, with an optional parameter to filter variants based on their minor allele frequencies. Additional heritability has been uncovered over the last decade with the increase in GWA study sample size. For example, a 2009 study with 3322 cases and 3,587 controls detected only a single genomic locus associated with schizophrenia[44], but by 2014, the number of genetic loci associated with schizophrenia had increased to 108 by using a sample size of over 36,000 cases and controls[70]. Although the number of variants identified have increased with GWA study sample size, the effect size for the majority of significant GWA loci is under 1.1, which makes it difficult to determine the individual functional effects of each identified variant[66]. A polygenic risk score confronts this matter by aggregating the individual effects of GWA study variants, but it also assumes that the genetic risk is additive.

The polygenic risk scores calculated for the individuals in the ADNI dataset reveal that the PRSKB is effective at estimating disease risk. As shown in Fig. 1, individuals with Alzheimer's disease had significantly higher genetic risk scores for Alzheimer's disease than individuals with mild cognitive impairment or who were cognitively normal. Recent findings by Leonenko, et al.[71] show that polygenic risk scores account for the severity of cognitive decline. Leonenko, et al.[71] demonstrated that the *APOE* gene was found to be the best predictor of amyloid deposition—a pathological hallmark of Alzheimer's disease and an important factor in neural degeneration. However, they also found that progression from amyloid accumulation and mild cognitive impairment to Alzheimer's disease was better determined by polygenic risk scores, not *APOE* status. Our polygenic risk score calculations similarly show that polygenic risk scores are effective at capturing the distinction between mild cognitive impairment and Alzheimer's disease in the ADNI cohort.

The analyses on the ADNI cohort also highlight the utility of polygenic risk scores in identifying groups of individuals with distinct genetic risk for a certain trait. For example, a Welch's two-sample t test revealed that genetic risk for B-Cell Acute Lymphoblastic Leukemia is significantly different between individuals with and without Alzheimer's disease (t = -9.3704; $P = 1.0631 \times 10^{-14}$), as shown in Supplementary Data 6. Ongoing studies involving the role of B cells in Alzheimer's disease show that B cell depletion counterintuitively decreases amyloid beta buildup in mice and may be a therapeutic target for Alzheimer's disease[72]. The PRSKB also identified a clear difference in genetic risk for insomnia in the Alzheimer's disease cohort(t = -7.9373; $P = 4.5937 \times 10^{-11}$), which is in-line with previous studies showing links between sleep patterns and Alzheimer's disease[73]. Our polygenic risk score analyses may help researchers to further examine other links between both known and unknown disease associations. By facilitating large-scale polygenic risk score analyses utilizing various genome-wide significant associations, we provide a tool to detect diseases with shared genetic bases that may lead to better risk analyses, cohort selection, and disease pathway analyses.

As GWA studies continue to improve, the polygenic risk score calculations computed in the PRSKB will become more powerful and effective. Recent efforts to recognize and improve the lack of diversity in GWA study sample populations[25,74] will allow users to compute polygenic risk scores for a wider range of ethnicities and help reduce population biases in polygenic risk score calculations. Furthermore, as GWA study sample sizes increase, additional loci with genome-wide association will be revealed, resulting in more comprehensive polygenic risk scores. Empirical evidence indicates that for each complex phenotype, there is a threshold sample size above which the rate of variant discovery increases dramatically[75]. Moreover, the detection of risk variants has yet to plateau for any trait[75], suggesting that as large cohorts become increasingly available, polygenic risk scores will become more robust and informative.

The PRSKB simplifies access to data required for polygenic risk score calculations. No other tool includes a centralized online database and command line interface that allow users to simultaneously query thousands of studies on their own data through both an online and command line interface. We anticipate that the PRSKB will enhance the role of polygenic risk scores in future genetic studies of complex disease and trait heritability by streamlining the process to calculate polygenic risk scores across various studies.

## Methods

**Data compilation**. The PRSKB integrates with the National Human Genome Research Institute-European Bioinformatics Institute (NHGRI-EBI) GWAS Catalog[76] to provide the most up-to-date and comprehensive list of GWA studies. The GWAS Catalog is a publicly available database of GWA study summary statistics that allows individual research labs to submit full summary statistic files. The PRSKB automatically downloads, prunes, and reformats study and association data from the GWAS Catalog using the gwasrapidd R library[77]. The data are filtered to include only associations that contain both a beta value (or odds ratio) and the respective risk allele. Each variant is analyzed independently (i.e., risk haplotypes are excluded). Sex-specific variants are not included in the database. Finally, any allele that has been reported on the reverse strand is automatically detected and flipped to the forward strand. The strand-flipping procedure entails comparing each reported risk allele to the list of possible alleles for the specified variant from dbSNP[78]. If the reported risk allele does not exist in the list of possible alleles, the

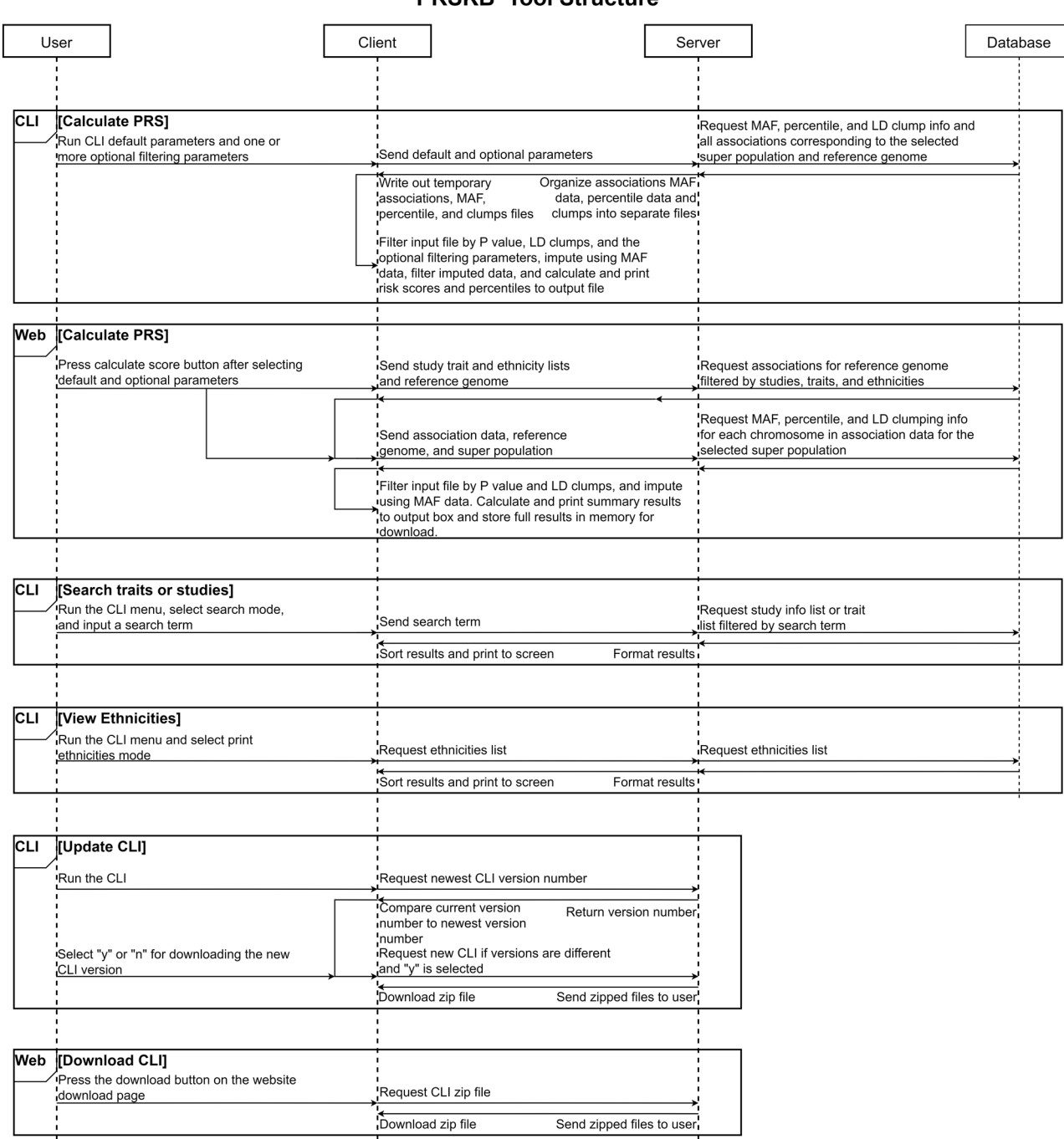

**Fig. 3 The PRSKB Tool Structure.** The PRSKB tool is composed of a client, a server, and a database. The user interacts with the client, which is either the web tool (https://prs.byu.edu), or the command-line interface (CLI). The client connects to the server that then retrieves and returns data from the PRSKB database to the client. The arrows in this diagram represent the flow of data. Boxes represent specific actions a PRSKB user can take with an icon indicating the client type for each box.

complement of the risk allele is checked against the dbSNP list. If the complement is present, then it is used as the reported risk allele for polygenic risk score calculations, as recommended by Choi, et al.[21]. Ambiguous variants that cannot be resolved by this method are automatically excluded from the analyses.

**PRSKB tool structure**. The PRSKB is divided into three key parts: the database, the server, and the client, as shown in Fig. 3. More information on how the database was compiled is shown in Supplementary Fig. 1. The GWA study data, linkage disequilibrium clumping data, and association data are housed in a MySQL database on the PRSKB server. Supplementary Tables 1–3 expound on the information found in each database table. The variant associations from each study/trait combination are contained within a single *associations table*, which includes

detailed summary statistics for each variant (see Supplementary Table 1). The *study table* (see Supplementary Table 2) contains detailed descriptions of each GWA study. Finally, there are four clumps tables, *hg38 clumps*, *hg19 clumps*, *hg18 clumps*, and *hg17 clumps*, that include linkage disequilibrium region identification numbers for variants in each of the five super populations from the 1000 Genomes project (see Supplementary Table 3). The associations and study tables are automatically updated monthly with new associations added to the GWAS Catalog. The scripts for loading tables into the database are publicly available at https://github.com/kauwelab/PolyRiskScore/tree/master/update_database_scripts.

The server houses the application programming interface endpoints for the PRSKB, running NodeJS using PM2 (https://pm2.keymetrics.io/) and NGINX (https://www.nginx.com/). While the user does not interact directly with the

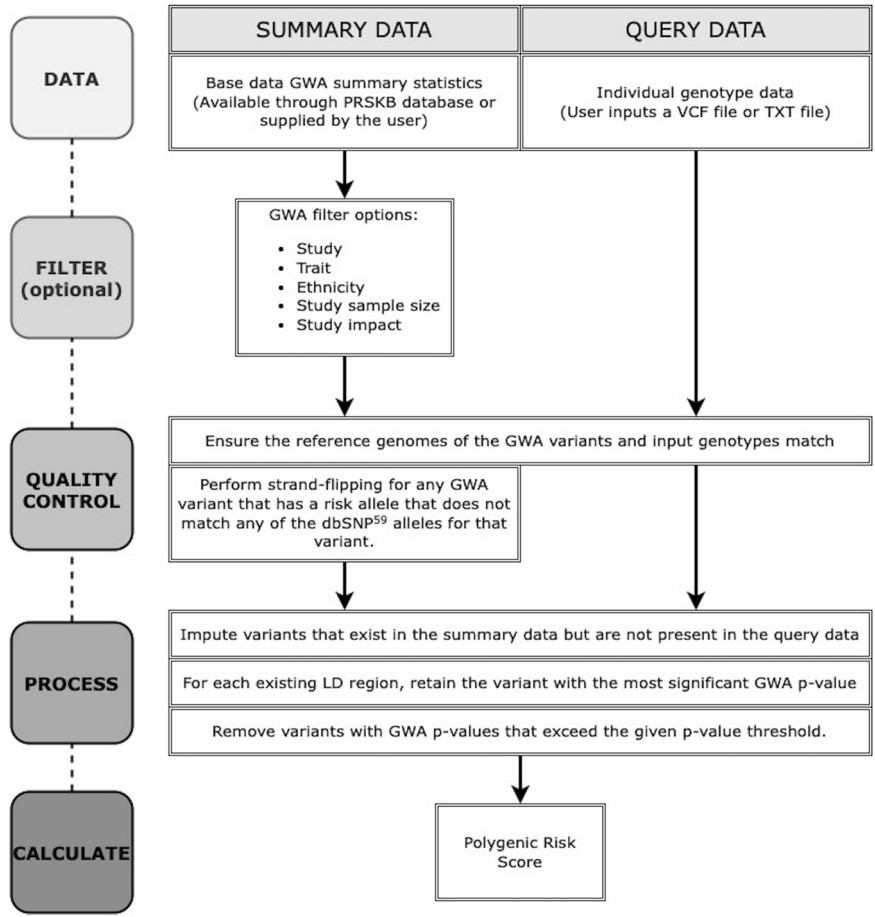

**Fig. 4 Polygenic risk score workflow.** The process follows the standards established by Choi et al.[21].

application programming interface endpoints, the client calls endpoints to download requested data needed to calculate polygenic risk scores. All calculations occur client-side to reduce strain on the server.

Users have two platforms from which they can calculate polygenic risk scores. The first platform is a web interface accessible at https://prs.byu.edu via a web browser that allows users to perform client-side calculations where user data are never uploaded to the PRSKB server. The second platform is a command-line interface (CLI) tool that can be run from the Linux or Mac command-line or from a bash shell on Windows. The CLI includes a bash script and four Python scripts. We recommend using the CLI to calculate polygenic risk scores for multi-sample VCF files, calculating scores spanning more than 50 GWA studies, and if the user prefers more control over their bioinformatics pipelines.

**Linkage disequilibrium clumping.** Linkage disequilibrium is the nonrandom association of alleles at two or more loci[79] and generally affects loci that reside in close physical proximity, resulting in the joint inheritance of alleles at different loci within families and populations. Genetic variants that are in high linkage disequilibrium will be similarly associated with traits in GWA studies. If they are not adequately assessed, they can confound a polygenic risk score analysis by over-representing the relative risk for a disease. For example, if three disease-associated loci are in high linkage disequilibrium, only one locus should be included in calculating a polygenic risk score because the same risk signal is present in any of those three loci.

Therefore, the genetic variants used to calculate polygenic risk scores need to be largely independent from each other to reduce score inflation. The PRSKB includes linkage disequilibrium values that were calculated by first separating the 1000 Genomes data into five previously-annotated superpopulations: African, American, East Asian, European, and South Asian. We then used PLINK Linkage Disequilibrium (LD) Clumping[80] to calculate linkage disequilibrium regions for the variants in each population. We ran this analysis for the data available in both reference genomes hg38 and hg19. Although linkage disequilibrium regions are nearly identical between reference genomes[81], we also converted the variant coordinates in each clump to reference genomes hg18 and hg17 so that user-supplied genotypes can be easily mapped to the correct LD clump regardless of reference genome.

The LD Clumping analysis results were subsequently used to assign each genetic variant to an LD clump identifier (clump ID) for each population. LD regions were

determined using an r-squared cutoff of 0.25 and a distance threshold of 500 kb, which correspond to parameters used in previous studies[64,65]. From this information, we created a table of population-specific linkage disequilibrium clusters for each reference genome in our database (see Supplementary Table 3). The clump ID for each population facilitates the dynamic retrieval of LD clumps from the database so that no more than one variant per LD region is included in an individual polygenic risk score calculation. Supplementary Fig. 2 illustrates the process used to account for linkage disequilibrium in the PRSKB calculations, and more information on how the clumps were created is found in Supplementary Note 1.

**Calculating polygenic risk scores.** Polygenic risk scores are calculated client-side, meaning no private data ever reaches our servers. The tool uses the same protocols outlined by Choi et al.[21]. Figure 4 shows that polygenic risk score calculations require two essential datasets: (1) summary data comprised of GWA study summary statistics (e.g., odds ratios or beta values, risk alleles, and p-values), and (2) user-supplied query data comprised of individual genotypes. Although a single GWA study is used to calculate each polygenic risk score, users can select multiple studies or traits, which will each be analyzed independently. Users can also use their own GWA summary statistics for personalized analyses. The PRSKB first ensures that the summary data and the query data are in the same format (e.g., strand flipping and same reference genome). Next, missing genotypes are imputed based on the minor allele frequency of either the sample or specified dataset (e.g., 1000 Genomes population or UK Biobank) and that frequency is used in the polygenic risk score calculation (e.g., if the minor allele frequency for a missing genotype were 0.2, then the reported risk attributed to that missing genotype would be 0.2 times 2 alleles times the associated risk from the GWA study). An optional parameter allows users to set an imputation threshold that removes studies from the output file where the number of imputed genotypes exceeds a specified percentage. By default, at least half of the genotypes used to calculate the polygenic risk score must be included in the sample. Linkage disequilibrium is then calculated by comparing each locus to the population-specific clumping regions for each GWA study that are housed on our server. When a sample has two or more variants within the same clumping region, the PRSKB chooses the variant with the most significant GWA p-value from that region to represent the clump in the polygenic risk score. The remaining set of independent variants is used in the polygenic risk score calculation. The PRSKB uses the simple additive model to calculate polygenic

risk scores by averaging the effects of all risk alleles across the genome. Missing variants are replaced with the population minor allele frequency of the risk allele in the same manner as PLINK[55] and PRSice-2[52]. We chose to implement this model because scores calculated using the additive model are generally highly accurate[11,21,26,29,82,83]. Although the additive polygenic risk score model does not account for gene-gene or gene-environment interactions, it facilitates comparisons with other available software. For example, the largest meta-analysis of heritability from twin studies validates the accuracy of a simple additive model for a majority of the traits examined[84].

**UK biobank and 1000 genomes polygenic risk score visualization.** In order to interpret polygenic risk scores, individual results must be contextualized against a large cohort of similar ethnicity[29]. The 1000 Genomes Project[58] contains the best representation of allele frequencies in unrelated individuals across diverse populations and has sequencing data for 2,504 unrelated individuals spanning five superpopulations. We also recognize that some users might want to contextualize their scores against a larger population. Therefore, we also included a separate cohort of 487,409 relatively healthy individuals of primarily European descent from the United Kingdom (UK) Biobank[59]. We used the PRSKB to compute polygenic risk scores from all GWA studies in our database for each individual in each cohort (each 1000 Genomes population was a different cohort). We then calculated the percentile rank of each person against all other people in the cohort. The polygenic risk score and percentile ranks were passed to Plotly JavaScript[85] to create interactive graphics that allow users to visualize population-specific distributions of polygenic risk scores for any study in the PRSKB database. Dynamic plots with a table of summary statistics for each study are available for users to query online at https://prs.byu.edu/visualize.html.

**Alzheimer's disease neuroimaging initiative (ADNI) case study.** We also computed Alzheimer's disease polygenic risk scores and interactive graphics for the Alzheimer's Disease Neuroimaging Initiative (ADNI) database (adni.loni.usc.edu) to verify the efficacy of the PRSKB calculations. ADNI was launched in 2003 as a public-private partnership, led by Principal Investigator Michael W. Weiner, MD. The primary goal of ADNI has been to test whether serial magnetic resonance imaging (MRI), positron emission tomography (PET), other biological markers, and clinical and neuropsychological assessment can be combined to measure the progression of mild cognitive impairment and early Alzheimer's disease. Mild cognitive impairment is the preclinical stage of Alzheimer's disease and is characterized by a slight but measurable decline in cognitive abilities. Individuals with mild cognitive impairment are at an increased risk of developing Alzheimer's disease or another dementia. All relevant ethical regulations were followed for establishing the ADNI cohort, including obtaining informed consent. All data were deidentified for our study, and we did not enroll any human participants.

We used all 808 whole-genome sequences from the ADNI cohort that also have a clinical dementia rating (CDR) score (see Supplementary Table 4 for the number of samples in each CDR group). Population structure was previously analyzed[86] and shows that the ADNI whole-genome sequencing participants are primarily similar to the European population in the 1000 Genomes Project. We recognize that uncorrected population structure can either inflate or deflate polygenic risk score associations when the population structure of the base and target samples significantly differ[21]. Inaccurate adjustments for population structure can also introduce biases into polygenic risk scores[21]. We decided not to correct for population structure in ADNI because (1) the population structure for the base data from the genome-wide association studies included in the GWAS Catalog indicate general geographic locations for the included subjects without including principal components, and (2) the principal component analysis of the ADNI whole genome sequences shows that the population structure of ADNI is largely similar to the general geographic location of the base data. Both the PRSKB and PRSice-2 were run using the same assumptions to ensure that the results are directly comparable.

CDR is a summary measure developed to denote the overall severity of dementia in an individual, where CDR = 0 is considered normal cognition, CDR = 0.5 is mild cognitive impairment, and CDR ≥ 1.0 is Alzheimer's disease[87]. As a case study, we used the PRSKB calculator to compute the polygenic risk scores for each ADNI participant for three Alzheimer's disease GWA studies available in our database: Lambert et al.[3], Jansen et al.[2], and Lo et al.[62]. The genetic variants used for each polygenic risk score calculation are listed in Supplementary Data 1–3. The PRSKB imputed missing genotypes using the entire ADNI cohort minor allele frequency and used variant linkage disequilibrium based on the European population in the 1000 Genomes Project.

A Kolmogorov-Smirnov test of normality[88] revealed that the risk scores were not normally distributed (Alzheimer's disease $P = 2.2 \times 10^{-16}$, mild cognitive impairment $P = 4.4 \times 10^{-16}$, cognitively normal $P = 2.2 \times 10^{-16}$), so we opted to use a Mann-Whitney U test[89] to compare the distributions of polygenic risk scores between individuals with and without Alzheimer's disease. We first compared genetic risk scores in individuals with a CDR ≥ 1 (Alzheimer's disease) to individuals with a CDR ≤ 0.5 (mild cognitive impairment + cognitively normal). Next, we compared individuals with a CDR = 0 (cognitively normal) to individuals with a CDR ≥ 0.5 (Alzheimer's disease + mild cognitive impairment). Those results were compared to similar calculations from another leading polygenic risk score calculator, PRSice-2[52], to assess the congruence

between the two algorithms as well as their ability to differentiate between the three cognitive groups in ADNI.

We performed similar analyses using each study and trait in the PRSKB database to identify additional diseases or traits that are not typically associated with Alzheimer's disease but might be covariates in the ADNI dataset or significantly correspond with CDR. We report two clustering comparisons: (1) Individuals with Alzheimer's disease (CDR ≥ 1.0) and all other individuals (CDR ≤ 0.5) and (2) Individuals with normal cognition (CDR = 0) and individuals with any cognitive impairment (CDR ≥ 0.5). We did not analyze mild cognitive impairment as a separate group to maintain statistical power. Similar to the computations performed with the UK Biobank and 1000 Genomes datasets, we also report the percentile score distributions and summary statistics for CDR ≥ 1, CDR = 0.5, and CDR = 0 online using Plotly Javascript[85].

**Reporting summary.** Further information on research design is available in the Nature Research Reporting Summary linked to this article.

## Data availability

This project is documented online at https://polyriskscore.readthedocs.io/en/latest/. A web interface is publicly available at https://prs.byu.edu/. All data and analyses are publicly available through the web interface and the GWAS Catalog (https://www.ebi.ac.uk/gwas/). Sequencing and participant data were not collected or generated for this study.

## Code availability

All programs and code for this manuscript are publicly available at https://github.com/kauwelab/PolyRiskScore and https://doi.org/10.5281/zenodo.6705589.

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

## Acknowledgements

This work was supported by the BrightFocus Foundation and its donors [A2020118F to Miller; A2020161S to Ebbert], the National Institutes of Health [RF1AG054052 to Kauwe; 1P30AG072946-01 to the University of Kentucky Alzheimer's Disease Research Center; AG068331 to Ebbert; GM138636 to Ebbert], and the Alzheimer's Association [2019-AARG-644082 to Ebbert]. We also acknowledge Brigham Young University and the University of Kentucky for supporting this research. Data collection and sharing for this project was funded by the Alzheimer's Disease Neuroimaging Initiative (ADNI) (National Institutes of Health Grant U01 AG024904) and DOD ADNI (Department of Defense award number W81XWH-12-2-0012). ADNI is funded by the National Institute on Aging, the National Institute of Biomedical Imaging and Bioengineering, and through generous contributions from the following: AbbVie, Alzheimer's Association; Alzheimer's Drug Discovery Foundation; Araclon Biotech; BioClinica, Inc.; Biogen; Bristol-Myers Squibb Company; CereSpir, Inc.; Cogstate; Eisai Inc.; Elan Pharmaceuticals, Inc.; Eli Lilly and Company; EuroImmun; F. Hoffmann-La Roche Ltd and its affiliated company Genentech, Inc.; Fujirebio; GE Healthcare; IXICO Ltd.; Janssen Alzheimer Immunotherapy Research & Development, LLC.; Johnson & Johnson Pharmaceutical Research & Development LLC.; Lumosity; Lundbeck; Merck & Co., Inc.; Meso Scale Diagnostics, LLC.; NeuroRx Research; Neurotrack Technologies; Novartis Pharmaceuticals Corporation; Pfizer Inc.; Piramal Imaging; Servier; Takeda Pharmaceutical Company; and Transition Therapeutics. The Canadian Institutes of Health Research is providing funds to support ADNI clinical sites in Canada. Private sector contributions are facilitated by the Foundation for the National Institutes of Health (www.fnih.org). The grantee organization is the Northern California Institute for Research and Education, and the study is coordinated by the Alzheimer's Therapeutic Research Institute at the University of Southern California. ADNI data are disseminated by the Laboratory for Neuro Imaging at the University of Southern California. Data used in preparation of this article were obtained from the Alzheimer's Disease Neuroimaging Initiative (ADNI) database (adni.loni.usc.edu). As such, the investigators within the ADNI contributed to the design and implementation of ADNI and/or provided data but did not participate in analysis or writing of this report. Per the ADNI data usage agreement, ADNI affiliations are listed in the Acknowledgements. A full list of ADNI investigators and their affiliations appears in the Acknowledgement List for ADNI Publications and at https://adni.loni.usc.edu/wp-content/uploads/how_to_apply/ADNI_Acknowledgement_List.pdf.

## Author contributions

All authors contributed to this work and approved the final version of this manuscript. MLP, ELV, and MEC should be regarded as co-first authors. They each contributed intellectually to the development of the PRSKB website and command line interface. They were primarily responsible for drafting the manuscript and online documentation, testing the PRSKB and associated methods, and organizing the workflows. ER and LD helped develop the online interface, drafted some sections of the manuscript, and tested the PRSKB. MTWE helped edit the manuscript, provided resources for its completion, and contributed intellectually by suggesting some additional features. The Alzheimer's Disease Neuroimaging Initiative provided genetic data that were used to validate the PRSKB. JBM and JSKK should be regarded as co-last authors. JBM and JSKK were the principal intellectual drivers of this tool, directed and mentored the other authors, provided resources to develop and deploy the PRSKB, wrote and revised the manuscript, and ensured the integrity of all analyses.

## Competing interests

The authors declare that there is a competing interest. J.B.M. and J.S.K.K. cofounded The BYU Genetic Risk Assessment and PolyScores Reports, which is a commercial venture that calculates polygenic risk scores from consumer DNA tests. All other authors declare no competing interests.

## Additional information

# The Alzheimer's Disease Neuroimaging Initiative
**Principal Investigator** M.W. Weiner[6]

## ATRI PI and Director of Coordinating Center Clinical Core P. Aisen[7] & R. Petersen[8]

## Executive Committee M. W. Weiner[6], P. Aisen[7], R. Petersen[8], C. R. Jack Jr[8], W. Jagust[9], J. Q. Trojanowki[10], A. W. Toga[7], L. Beckett[11], R. C. Green[12], A. J. Saykin[13], J. C. Morris[14], R. J. Perrin[14] & L. M. Shaw[10]

**ADNI External Advisory Board (ESAB)** Z. Khachaturian[15], M. Carrillo[16], W. Potter[17], L. Barnes[18], M. Bernard[19], H. González[20], C. Ho[21], J. K. Hsiao[22], J. Jackson[23], E. Masliah[19], D. Masterman[24], O. Okonkwo[25], R. Perrin[14], L. Ryan[19] & N. Silverberg[19]

**ADNI 3 Private Partner Scientific Board (PPSB)** A. Fleisher[26]

**Administrative Core - Northern California Institute for Research & Education (NCIRE / The Veterans Health Research Institute)** M. W. Weiner[6], D. T. Sacrey[27], J. Fockler[6], C. Conti[27], D. Veitch[27], J. Neuhaus[6], C. Jin[6], R. Nosheny[6], M. Ashford[27], D. Flenniken[27] & A. Kormos[27]

**Data and Publications Committee** R. C. Green[12]

**Resource Allocation Review Committee** T. Monine[28] & C. Conti[27]

**Clinical Core Leaders and Key Personnel** R. Petersen[8], P. Aisen[7], M. Rafii[7], R. Raman[7], G. Jimenez[7], M. Donohue[7], D. Gessert[7], J. Salazar[7], C. Zimmerman[7], Y. Cabrera[7], S. Walter[7], G. Miller[7], G. Coker[7], T. Clanton[7], L. Hergesheimer[7], S. Smith[7], O. Adegoke[7], P. Mahboubi[7], S. Moore[7], J. Pizzola[7], E. Shaffer[7] & B. Sloan[7]

**Biostatistics Core Leaders and Key Personnel** L. Beckett[11], D. Harvey[11] & M. Donohue[7]

**MRI Core Leaders and Key Personnel** C. R. Jack Jr[8], A. Forghanian-Arani[8], B. Borowski[8], C. Ward[8], C. Schwarz[8], D. Jones[8], J. Gunter[8], K. Kantarci[8], M. Senjem[8], P. Vemuri[8], R. Reid[8], N. C. Fox[29], I. Malone[29], P. Thompson[30], S. I. Thomopoulos[30], T. M. Nir[30], N. Jahanshad[30], C. DeCarli[11], A. Knaack[11], E. Fletcher[11], D. Harvey[11], D. Tosun-Turgut[6], S. R. Chen[27], M. Choe[27], K. Crawford[30], P. A. Yushkevich[10] & S. Das[10]

**PET Core Leaders and Key Personnel** W. Jagust[9], R. A. Koeppe[31], E. M. Reiman[32], K. Chen[32], C. Mathis[33] & S. Landau[9]

**Neuropathology Core Leaders and Key Personnel** J. C. Morris[14], R. Perrin[14], N. J. Cairns[14], E. Householder[14], E. Franklin[14], H. Bernhardt[14] & L. Taylor-Reinwald[14]

**Biomarkers Core Leaders and Key Personnel** L. M. Shaw[34], J. Q. Tojanowki[34], M. Korecka[34] & M. Figurski[34]

**Informatics Core Leaders and Key Personnel** A. W. Toga[7], K. Crawford[7] & S. Neu[7]

**Genetics Core Leaders and Key Personnel** A. J. Saykin[13], K. Nho[13], S. L. Risacher[13], L. G. Apostolova[13], L. Shen[34], T. M. Foroud[13], K. Nudelman[13], K. Faber[13] & K. Wilmes[13]

**Initial Concept Planning and Development** M. W. Winer[6], L. Thal[20] & Z. Khachaturian[15]

**National Institute on Aging** J. K. Hsiao[19]

**Oregon Health & Science University Investigators** L. C. Silbert[35], B. Lind[35], R. Crissey[35], J. A. Kaye[35], R. Carter[35], S. Dolen[35] & J. Quinn[35]

**University of Southern California Investigators** L. S. Schneider[7], S. Pawluczyk[7], M. Becerra[7], L. Teodoro[7], K. Dagerman[7] & B. M. Spann[7]

**University of California, San Diego Investigators** J. Brewer[20], H. Vanderswag[20] & A. Fleisher[20]

**University of Michigan Investigators** J. Ziolkowski[31], J. L. Heidebrink[31], L. Zbizek-Nulph[31] & J. L. Lord[31]

**Mayo Clinic, Rochester Investigators** R. Petersen[8], S. S. Mason[8], C. S. Albers[8], D. Knopman[8] & K. Johnson[8]

**Baylor College of Medicine Investigators** J. Villanueva-Meyer[36], V. Pavlik[36], N. Pacini[36], A. Lamb[36], J. S. Kass[36], R. S. Doody[36], V. Shibley[36], M. Chowdhury[36], S. Rountree[36] & M. Dang[36]

**Columbia University Medical Center Investigators** Y. Stern[37], L. S. Honig[37] & A. Mintz[37]

**Washington University in St. Louis Investigators** B. Ances[14], J. C. Morris[14], D. Winkfield[14], M. Carroll[14], G. Stobbs-Cucchi[14], A. Oliver[14], M. L. Creech[14], M. A. Mintun[14] & S. Schneider[14]

**University of Alabama, Birmingham Investigators** D. Geldmacher[38], M. N. Love[38], R. Griffith[38], D. Clark[38], J. Brockington[38] & D. Marson[38]

**Mount Sinai School of Medicine Investigators** H. Grossman[39], M. A. Goldstein[39], J. Greenberg[39] & E. Mitsis[39]

**Rush University Medical Center Investigators** R. C. Shah[18], M. Lamar[18] & P. Samuels[18]

**Wien Center Investigators** R. Duara[40], M. T. Greig-Custo[40] & R. Rodriguez[40]

**Johns Hopkins University Investigators** M. Albert[41], C. Onyike[41], L. Farrington[41], S. Rudow[41], R. Brichko[41] & S. Kielb[41]

**University of South Florida: USF Health Byrd Alzheimer's Institute Investigators** A. Smith[42], B. A. Raj[42] & K. Fargher[42]

**New York University Investigators** M. Sadowski[43], T. Wisniewski[43], M. Shulman[43], A. Faustin[43], J. Rao[43], K. M. Castro[43], A. Ulysse[43], S. Chen[43], M. O. Sheikh[43] & J. Singleton-Garvin[43]

**Duke University Medical Center Investigators** P. M. Doraiswamy[44], J. R. Petrella[44], O. James[44], T. Z. Wong[44] & S. Borges-Neto[44]

**University of Pennsylvania Investigators** J. H. Karlawish[10], D. A. Wolk[10], S. Vaishnavi[10], C. M. Clark[10] & S. E. Arnold[10]

**University of Kentucky Investigators** C. D. Smith[1], G. A. Jicha[1], R. E. Khouli[1] & F. D. Raslau[1]

**University of Pittsburgh Investigators** O. L. Lopez[33], M. Oakley[33] & D. M. Simpson[33]

**University of Rochester Medical Center Investigators** A. P. Porsteinsson[45], K. Martin[45], N. Kowalski[45], M. Keltz[45], B. S. Goldstein[45], K. M. Makino[45], M. S. Ismail[45] & C. Brand[45]

**University of California Irvine Institute for Memory Impairments and Neurological Disorders Investigators**
G. Thai[46], A. Pierce[46], B. Yanez[46], E. Sosa[46] & M. Witbracht[46]

**University of Texas Southwestern Medical School Investigators** B. Kelley[47], T. Nguyen[47], K. Womack[47], D. Mathews[47] & M. Quiceno[47]

**Emory University Investigators** A. I. Levey[48], J. J. Lah[48], I. Hajjar[48] & J. S. Cellar[48]

**University of Kansas Medical Center Investigators** J. M. Burns[49], R. H. Swerdlow[49] & W. M. Brooks[49]

**University of California, Los Angeles Investigators** D. H. S. Silverman[50], S. Kremen[50], L. Apostolova[50], K. Tingus[50], P. H. Lu[50], G. Bartzokis[50], E. Woo[50] & E. Teng[50]

**Mayo Clinic, Jacksonville Investigators** N. R. Graff-Radford[51], F. Parfitt[51] & K. Poki-Walker[51]

**Indiana University Investigators** M. R. Farlow[13], A. M. Hake[13], B. R. Matthews[13], J. R. Brosch[13] & S. Herring[13]

**Yale University School of Medicine Investigators** C. H. van Dyck[52], A. P. Mecca[52], S. P. Good[52], M. G. MacAvoy[52], R. E. Carson[52] & P. Varma[52]

**McGill University, Montreal-Jewish General Hospital Investigators** H. Chertkow[53], S. Vaitekunis[53] & C. Hosein[53]

**Sunnybrook Health Sciences, Ontario Investigators** S. Black[54], B. Stefanovic[54] & C. Heyn[54]

**University of British Columbia Clinic for Alzheimer's Disease and Related Disorders Investigators**
G. R. Hsiung[55], E. Kim[55], B. Mudge[55], V. Sossi[55], H. Feldman[55] & M. Assaly[55]

**St. Joseph's Health Care Investigators** E. Finger[56], S. Pasternak[56], I. Rachinsky[56], A. Kertesz[56], D. Drost[56] & J. Rogers[56]

**Northwestern University Investigators** I. Grant[57], B. Muse[57], E. Rogalski[57], J. Robson[57], M. Mesulam[57], D. Kerwin[57], C. Wu[57], N. Johnson[57], K. Lipowski[57], S. Weintraub[57] & B. Bonakdarpour[57]

**Nathan Kline Institute Investigators** N. Pomara[58], R. Hernando[58] & A. Sarrael[58]

**University of California, San Francisco Investigators** H. J. Rosen[6], B. L. Miller[6] & D. Perry[6]

**Georgetown University Medical Center Investigators** R. S. Turner[59], K. Johnson[59], B. Reynolds[59], K. McCann[59] & J. Poe[59]

**Brigham and Women's Hospital Investigators** R. A. Sperling[12], K. A. Johnson[12] & G. A. Marshall[12]

**Stanford University Investigators** J. Yesavage[60], J. L. Taylor[60], S. Chao[60], J. Coleman[60], J. D. White[60], B. Lane[60], A. Rosen[60] & J. Tinklenberg[60]

**Banner Sun Health Research Institute Investigators** C. M. Belden[61], A. Atri[61], B. M. Spann[61], K. A. Clark[61], E. Zamrini[61] & M. Sabbagh[61]

**Boston University Investigators** R. Killiany[62], R. Stern[62], J. Mez[62], N. Kowall[62] & A. E. Budson[62]

**Howard University Investigators** T. O. Obisesan[63], O. E. Ntekim[63], S. Wolday[63], J. I. Khan[63], E. Nwulia[63] & S. Nadarajah[63]

**Case Western Reserve University Investigators** A. Lerner[64], P. Ogrocki[64], C. Tatsuoka[64] & P. Fatica[64]

**University of California, Davis-Sacramento Investigators** E. Fletcher[65], P. Maillard[65], J. Olichney[65], C. DeCarli[65] & O. Carmichael[65]

**Dent Neurologic Institute Investigators** V. Bates[66], H. Capote[66] & M. Rainka[66]

**Parkwood Institute Investigators** M. Borrie[67], T. Lee[67] & R. Bartha[67]

**University of Wisconsin Investigators** S. Johnson[25], S. Asthana[25] & C. M. Carlson[25]

**Banner Alzheimer's Institute Investigators** A. Perrin[32] & A. Burke[32]

**Ohio State University Investigators** D. W. Scharre[68], M. Kataki[68], R. Tarawneh[68] & B. Kelley[68]

**Albany Medical College Investigators** D. Hart[69], E. A. Zimmerman[69] & D. Celmins[69]

**University of Iowa College of Medicine Investigators** D. D. Miller[70], L. L. B. Ponto[70], K. E. Smith[70], H. Koleva[70], H. Shim[70], K. W. Nam[70] & S. K. Schultz[70]

**Wake Forest University Health Sciences Investigators** J. D. Williamson[71], S. Craft[71], J. Cleveland[71], M. Yang[71] & K. M. Sink[71]

**Rhode Island Hospital Investigators** B. R. Ott[72], J. D. Drake[72], G. Tremont[72] & L. A. Daiello[72]

**Cleveland Clinic Lou Ruvo Center for Brain Health Investigators** M. Sabbagh[73], A. Ritter[73], C. Bernick[73], D. Munic[73] & A. Mintz[73]

**Roper St. Francis Healthcare Investigators** A. O'Connell[74], J. Mintzer[74] & A. Williams[74]

**Houston Methodist Neurological Institute Investigators** J. Masdeu[75]

**Barrow Neurological Institute Investigators** J. Shi[76], A. Garcia[76] & M. Sabbagh[76]

**Vanderbilt University Medical Center Investigators** P. Newhouse[77]

**Long Beach Veterans Affairs Neuropsychiatric Research Program Investigators** S. Potkin[78]

**Butler Hospital Memory and Aging Program Investigators** S. Salloway[79], P. Malloy[79] & S. Correia[79]

**Neurological Care of Central New York Investigators** S. Kittur[80]

**Hartford Hospital, Olin Neuropsychiatry Research Center Investigators** G. D. Perlson[81], K. Blank[81] & K. Anderson[81]

**Dartmouth-Hitchcock Medical Center Investigators** L. A. Flashman[82], M. Seltzer[82], M. L. Hynes[82] & R. B. Santulli[82]

**Cornell University Investigators** N. Relkin[83], G. Chiang[83], A. Lee[83], M. Lin[83] & L. Ravdin[83]

[6]University of California, San Francisco, San Francisco, CA, USA. [7]University of Southern California, Los Angeles, CA, USA. [8]Mayo Clinic, Rochester, Rochester, MN, USA. [9]University of California, Berkeley, Berkeley, CA, USA. [10]University of Pennsylvania, Philadelphia, PA, USA. [11]University of California, Davis, Davis, CA, USA. [12]Brigham and Women's Hospital, Harvard Medical School, Boston, MA, USA. [13]Indiana University, Bloomington, IN, USA. [14]Washington University in St. Louis, St. Louis, MO, USA. [15]Prevent Alzheimer's Disease, Rockville, MD, USA. [16]Alzheimer's Association, Chicago, IL, USA. [17]National Institute of Mental Health, Bethesda, MD, USA. [18]Rush University, Chicago, IL, USA. [19]National Institute on Aging, Bethesda, MD, USA. [20]University of California, San Diego, San Diego, CA, USA. [21]Denali Therapeutics, South San Francisco, CA, USA. [22]National Institutes of Health, Bethesda, MD, USA. [23]Massachusetts General Hospital, Boston, MA, USA. [24]Biogen, Cambridge, MA, USA. [25]University of Wisconsin, Madison, Madison, WI, USA. [26]Eli Lilly, Indianapolis, IN, USA. [27]The Veterans Health Research Institute, Northern California Institute for Research and Education, San Francisco, CA, USA. [28]University of Washington, Seattle, WA, USA. [29]University College London, London, GB, USA. [30]University of Southern California School of Medicine, Los Angeles, CA, USA. [31]University of Michigan, Ann Arbor, MI, USA. [32]Banner Alzheimer's Institute, Phoenix, AZ, USA. [33]University of Pittsburgh, Pittsburgh, PA, USA. [34]Perelman School of Medicine, University of Pennsylvania, Philadelphia, PA, USA. [35]Oregon Health and Science University, Portland, OR, USA. [36]Baylor College of Medicine, Houston, TX, USA. [37]Columbia University Medical Center, New York City, NY, USA. [38]University of Alabama, Birmingham, Birmingham, AL, USA. [39]Mount Sinai School of Medicine, New York City, NY, USA. [40]Wien Center for Alzheimer's Disease and Memory Disorders, Miami, FL, USA. [41]Johns Hopkins University, Baltimore, MD, USA. [42]University of South Florida: Health Byrd Alzheimer's Institute, Tampa, FL, USA. [43]New York University, New York City, NY, USA. [44]Duke University Medical Center, Durham, NC, USA. [45]University of Rochester Medical Center, Rochester, NY, USA. [46]University of California Irvine Institute for Memory Impairments and Neurological Disorders, Irvine, CA, USA. [47]University of Texas Southwestern Medical School, Dallas, TX, USA. [48]Emory University, Atlanta, GA, USA. [49]University of Kansas Medical Center, Kansas City, KS, USA. [50]University of California, Los Angeles, Los Angeles, CA, USA. [51]Mayo Clinic, Jacksonville, Jacksonville, FL, USA. [52]Yale University School of Medicine, New Haven, CT, USA. [53]McGill University, Montreal-Jewish General Hospital, Montreal, Quebec, Canada. [54]Sunnybrook Health Sciences, Ontario, Toronto, Ontario, Canada. [55]University of British Columbia Clinic for Alzheimer's Disease and Related Disorders, Vancouver, British Columbia, Canada. [56]St. Joseph's Health Care, London, Ontario, Canada. [57]Northwestern University, Evanston, IL, USA. [58]Nathan Kline Institute, Orangeburg, NY, USA. [59]Georgetown University Medical Center, Washington, D.C. USA. [60]Stanford University, Stanford, CA, USA. [61]Banner Sun Health Research Institute, Sun City, AZ, USA. [62]Boston University, Boston, MA, USA. [63]Howard University, Washington, D.C, USA. [64]Case Western Reserve University, Cleveland, OH, USA. [65]University of California, Davis-Sacramento, Sacramento, CA, USA. [66]Dent Neurologic Institute, Amherst, NY, USA. [67]Parkwood Institute, London, Ontario, Canada. [68]Ohio State University, Columbus, OH, USA. [69]Albany Medical College, Albany, NY, USA. [70]University of Iowa College of Medicine, Iowa City, IA, USA. [71]Wake Forest University Health Sciences, Winston-Salem, NC, USA. [72]Rhode Island Hospital, Providence, RI, USA. [73]Cleveland Clinic Lou Ruvo Center for Brain Health, Las Vegas, NV, USA. [74]Roper St. Francis Healthcare, Charleston, SC, USA. [75]Houston Methodist Neurological Institute, Houston, TX, USA. [76]Barrow Neurological Institute, Phoenix, AZ, USA. [77]Vanderbilt University Medical Center, Nashville, TN, USA. [78]Long Beach Veterans Affairs Neuropsychiatric Research Program, Long Beach, CA, USA. [79]Butler Hospital Memory and Aging Program, Warren Alpert Medical School, Brown University, Providence, RI, USA. [80]Neurological Care of Central New York, Liverpool, NY, USA. [81]Hartford Hospital, Olin Neuropsychiatry Research Center, Hartford, CT, USA. [82]Dartmouth-Hitchcock Medical Center, Lebanon, NH, USA. [83]Cornell University, Ithaca, NY, USA.

