## [Peer Review File · Communications Biology]

Reviewers' comments:

Reviewer #1 (Remarks to the Author):

The Polygenic Risk Score Knowledge Base: A centralized online repository for calculating and contextualizing polygenic risk scores
Madeline L. Page et al.

The authors introduce a useful resource, Polygenic Risk Score Knowledge Base (PRSKB), which can estimate polygenic risk scores based on > 2,300 GWAS data. The PRSKB is web-based interface through which users can upload their GWAS data. As an example, they report study-specific polygenic risk scores using several datasets including UK Biobank data. The authors conclude that the PRSKB will enhance the role of polygenic risk scores (PRS) in the complex disease analyses.

It is impressive with the number of GWAS data available in PRSKB (> 2,300). Although I agree with the authors that the PRSKB will be useful for scientists in the field of PRS research, I have several comments and questions to improve the current version of manuscript.

1. How about the accuracy of PRSs estimated from PRSKB calculator? Have the authors compared the accuracy with other existing state-of-the-art methods? The authors could have added such results (comparison) in Results section, which will help convincing users why they should use PRSKB.

2. Have the authors checked the computing speed of PRSKB calculator, compared to other existing methods/software? For example, if a user wants to upload 500,000 samples genotyped for 40M SNPs, how long would it take to get PRS? The authors should make this clearer in their manuscript (in a table or figure) so that readers/users should be well informed.

3. Does it provide reliability of each PRS? It may be possible that PRSKB will be used to predict the genetic risk of an individual (clinical practice). In this case, the reliability of PRS should be provided.

4. Although a large number of GWAS data are available in PRSKB, they are already in public domain. So, the merit of PRSKB is to collect the large amount of information in one place, which make it easier for users to find reference data (GWAS summary stats). This can be much more useful if users can search relevant data according to their specific purpose, e.g. genotype-by-environment summary stats (please see GIANT consortium homepage). Please also note that a recent paper has introduced a method utilizing genotype-by-environment summary stats.

GxEsum: a novel approach to estimate the phenotypic variance explained by genome-wide GxE interaction based on GWAS summary statistics for biobank-scale data (2021) Genome Biology 22: 183.

5. There are a number of interfaces that are similar to PRSKB, e.g. Impute.me, Polygenic Score Catalog and so on. The authors should make it clearer what is novel and unique property of PRSKB, compared to those existing resources.

I sign my name.

S. Hong Lee

Reviewer #2 (Remarks to the Author):

The authors have developed an online service called PRSKB for calculating polygenic scores using a wide range of GWAS and uploaded target sample data, requiring just a few optional parameters for the user to select. PRSKB uses the p-value thresholding and clumping approach when calculating the polygenic scores, and scales the target sample PRS according to the distribution of PRS found

in a range of reference samples. To demonstrate the utility of PRSKB, the authors use PRSKB to calculate Alzheimer's disease PRS in the ADNI cohort, and then test for genetic associations with Alzheimer's disease onset and mild cognitive decline.

I commend the authors on PRSKB. There are many aspects which I think are great, and I think a resource like PRSKB will be useful for researchers. However, I have a number of concerns that should be addressed.

Major:

- The LD-clumping process seems unconventional and requires further clarification and perhaps alterations.

- o LD-clumping is typically performed only considering the variants intersecting the GWAS and target data, identifying a list of independent lead variants in the GWAS that are also available in the target sample. In contrast, PRSKB identifies broadly independent 'LD regions' across all variants within a given reference sample, selects one variant within each LD region based on the minimum p-value in the GWAS, and then identifies the variants overlapping with the target sample. I expect this approach will generate suboptimal polygenic scores for two reasons.

1. Retaining only one variant within each LD region may lead to independent genetic effects being excluded due to the lead variants being correlated with other SNPs in common, but uncorrelated with one another. Have the authors considered performing clumping separately for each GWAS?

2. By performing clumping and selecting lead variants without considering the variants available in the target sample (as indicated by line 211, and figure 2), the variants considered for polygenic scoring may not be present in the target sample, leading to a reduced performance of the polygenic scores. I appreciate that performing clumping specific to the intersect with the target sample is not as practical, and possibly considering the same SNPs for all target sample is advantageous, as the polygenic scores are more comparable across target samples. However, have the authors considered reducing the likelihood of missing data by restricting their polygenic scores to variants that are common and typically available after imputation, such as HapMap3 variants?

- o Given my previous comment regarding the increased likelihood of missing variants in the target sample, please can the authors describe how PRSKB handles missing variants when calculating the polygenic scores? A commonly used approach is to use the reference sample allele frequency to impute missing variants. However, to ensure the PRS are of good quality, there should also be an initial check that most variants considered are available in the target sample. Are any such checks provided?

- o Line 316: "The user must also indicate the population of the samples to perform accurate linkage disequilibrium clumping". It appears the LD clumping is based on the ancestry of the target sample. However, it is more appropriate to select the LD clumping reference based on the ancestry of the GWAS sample, as the LD reference aims to recapitulate the LD structure within the GWAS sample to avoid double counting non-independent genetic effects. Can the authors clarify if this is not the case, explain their reasoning if it is the case, or change their methodology to use an LD reference that matches the ancestry of the GWAS sample?

- It seems as though the genetic data of the target sample must be somehow uploaded to the PRSKB server. However, this is often not possible due to data privacy concerns. Or is it possible to download the reference data and carry out of the analysis locally, avoiding sending the data to the PRSKB server? Please can the authors clarify how data privacy concerns are considered, or acknowledge that this is a limitation of their service?

- The authors demonstrate how the PRSKB server can be used to generate polygenic scores for research using the ADNI sample. The authors report significant associations and argue that this validates their approach. Whilst this is true to some extent, I think it is important to compare the variance explained by polygenic scores derived using PRSKB with those derived using more conventional approaches, such as PRSice. If the variance explained is similar this would be reassuring that the PRSKB methodology is producing polygenic scores of comparable quality to those based on current practice.

Minor:

- Line 76: "polygenic risk scores can capture missing heritability". Can the authors please elaborate on this statement?

- Line 85: "nominally significant differences in genetic risk scores for bipolar disorder between patients with and without psychotic symptoms suggest a more valid subclassification of the disease". More valid than what? For a nominally significant association, I thought this statement

sounded quite strong.

- Line 99: "PRSice-2 requires extensive startup time to fully utilize all available options". Can the authors please explain what they mean by this? Is 'startup time' the time taken for someone to understand the software?
- Line 126: "The data are filtered to include only associations that contain both an odds ratio and the respective risk allele". Requiring an odds ratio implies PRSKB only include PRS for binary outcomes. Is this the case? If this is a limitation and should clearly highlighted elsewhere.
- Line 128: "any allele that has been reported on the reverse strand is automatically detected and flipped". Can you clarify whether non-synonymous variants are excluded, and if not, how non-synonymous SNPs are aligned?
- The website shows that users can upload GWAS summary statistics of their own, but I cannot see where this is described in the paper.
- Figure 1. The figure is useful but unfortunately the text is very small. I suggest the authors try to increase the font size.
- Line 208: "When an individual in the target data has two or more variants within the same clumping region, the PRSKB chooses the variant with the most significant p-value". This sentence makes it sounds variant selection does consider the variants available in the target sample, in contrast to descriptions in line 211 and Figure 2. Please can the authors clarify. Furthermore, it sounds like variant selection is done at the individual level, rather than the sample level. Presumably this is performed at the sample level, otherwise different variants will be considered for each individual, reducing the comparability of the PRS across individuals. Please can the authors clarify.
- Line 234: "we divided the samples by population region". I can see how this was done for the 1000 genomes sample since it has reliable ancestry data, but how was this achieved for ADNI and UK Biobank? Self-reported ancestry, or reference-projected principal components?
- Line 257: "collective significant relationship between age, sex, and polygenic risk score". I assume this means the regression was $PRS \sim Age + Sex$, but please can the authors clarify.
- Line 259: "polygenic risk score outcomes". What is a polygenic score outcome? Are you simply saying there was no association between polygenic scores and sex or age? Please clarify.
- Lines 313-315: "Next, the user must specify the reference genome (hg38, hg19, hg18, or hg17) used to sequence the input variants so that the associations queried from the database correspond to the same reference assembly". Presumably this is only necessary when using VCF format, since RSIDs are not specific to genome build? Please clarify.
- Line 319: "the largest study cohort measured by the initial sample size plus the replication sample size". Can the authors please clarify why it is useful to filter GWAS by the discovery + replication sample size? It seems more appropriate to filter GWAS by the sample size in the GWAS summary statistics available, which often is does not include the replication sample.
- The authors test for PRS associations with Alzheimer's disease phenotypes in part to demonstrate PRSKB works, but they also use the results for some inference about the aetiology of Alzheimer's disease. However, it does not appear the authors have accounted for population stratification when performing these analyses, by using principal components of ancestry as covariates, or linear mixed models. Can the authors highlight this as a limitation or account for ancestry as is standard when performing polygenic score analysis for inference?
- Line 450: "First, while the PRSKB contains over 17,900 GWA variants, haplotype associations, or associations that include multiple variants for a single effect size, were removed from the PRSKB database". I think this sentence could be clarified by making a clearer distinction between what is and isn't included PRSKB. For example "...contains over 17,900 GWA variants, we removed haplotype associations..."

Reviewer #3 (Remarks to the Author):

This manuscript describes the Polygenic Risk Score Knowledge Base (PRSKB), an online database and server for calculating polygenic scores from a large number of GWAS summary statistics originating the GWAS Catalog.

Overall, the manuscript is well written and represents a very large effort in collating and building this resource. This is a valuable contribution to the human genetics research community.

I have several comments/questions to help clarify the analysis and how the tool works.

Major comments:

* Even though the ADNI Alzheimer's analysis is simply an example and not the main focus of the manuscript, more information is needed:

- Some information about the ADNI study design (a table would be useful): how many individuals, sex, number of cases/controls, ages, etc.

- More details about the Alzheimer's PRS are needed: how many SNPs were used, was APOE4 included?

- Based on Fig 3, the PRS distribution was not substantially deviating from normality but there were some small noticeable deviations, particularly for individuals with CDR \geq 1. It could be due to the ascertainment bias of ADNI (case/control), population stratification, batch effects, or the strong effect of APOE4 alleles relative to the rest of the SNPs in the score. What does the PCA plot of ADNI look like, and were analyses adjusted for principal components / batch effects (if there were any)?

* It's not clear, is the option to show percentiles for UK Biobank or other datasets available also for user uploaded PRS or only for the precomputed PRS?

* Do the UK Biobank analyses only include the British whites or are all participants included?

* Is there a filter for minor allele frequency to ensure that only common variants are included in the PRS? Some GWAS include rare variants as well which could end up in the calculation.

Minor comments:

* Line 46, 'Over 71,000 genetic loci are currently implicated in diseases or traits' is not quite right, these are pairs of associations, not loci (each locus can be associated with multiple traits)

* Line 54 'However, a major limitation of GWA studies is that each genetic locus is evaluated individually. Polygenic risk scores address this issue by aggregating all effects', it's not quite right. Many/most PRS are based on GWAS, it's just that the goal is different: GWAS is focused on discovery of trait associations, PRS is about quantifying the total genetic liability towards a specific trait/disease (as discussed later).

* 'For instance, a study of hypertrophic cardiomyopathy in African Americans misclassified benign genetic variants as pathogenic, which could have been avoided if even a small number of African Americans were included in the control group' -> is this from a study of common variants or rare variants? Typically common variants are not labelled 'pathogenic'; a different example could be useful here since the PRS is mainly based on common variants.

* The PGS Catalog (<https://pgscatalog.org>) should be mentioned somewhere, it is highly relevant too.

- Line 127 'The data are filtered to include only associations that contain both an odds ratio and the respective risk allele.' -> I assume PRSKB is not limited only to case/control but can do continuous traits too, in which case it wouldn't be an odd ratio.

- 'Additionally, only non-haplotype associations that reside on an autosomal' -> what does 'haplotype' mean here, HLA alleles?

- I suggest replacing 'gender' with 'sex' (eg Line 457) since for the most part we are dealing with biological sex effects and not gender per se.

Dear Dr. Inglis,

We appreciate the positive and insightful reviews for our manuscript, *The Polygenic Risk Score Knowledge Base: A centralized online repository for calculating and contextualizing polygenic risk scores*. We address each comment below and revised the manuscript to reflect these changes.

Responses to Reviewer #1:

Reviewer Overview

The authors introduce a useful resource, Polygenic Risk Score Knowledge Base (PRSKB), which can estimate polygenic risk scores based on > 2,300 GWAS data. The PRSKB is a web-based interface through which users can upload their GWAS data. As an example, they report study-specific polygenic risk scores using several datasets including UK Biobank data. The authors conclude that the PRSKB will enhance the role of polygenic risk scores (PRS) in the complex disease analyses. It is impressive with the number of GWAS data available in PRSKB (> 2,300). Although I agree with the authors that the PRSKB will be useful for scientists in the field of PRS research, I have several comments and questions to improve the current version of manuscript.

- 1. How about the accuracy of PRSs estimated from PRSKB calculator? Have the authors compared the accuracy with other existing state-of-the-art methods? The authors could have added such results (comparison) in Results section, which will help convincing users why they should use PRSKB.**

We added the following comparison to PRSice-2 in addition to our case study on Alzheimer's disease (line 432):

Comparison to PRSice-2

The PRSKB reports similar polygenic risk score results as PRSice-2. Figure 5A plots the polygenic risk scores calculated for both the PRSKB and PRSice-2 across ADNI participants using the Lambert, et al. ³ GWA study. Since polygenic risk scores are a relative measurement of genetic risk compared to a population, we compared the shape of the distributions from the PRSKB and PRSice-2 to ensure that both algorithms report similar score distributions. After performing a minor transformation to have the same median values for both algorithms (original difference between medians is 0.001306), a Welch's two sample t-test shows that slight variations between the two algorithms do not change the overall shape of the distributions (see Figure 5B; $t=0.004782$; $P=0.9962$). Similar comparisons between Alzheimer's disease and cognitive normal controls in the ADNI dataset using GWA studies from Lambert, et al. ³, Jansen, et al. ², and Lo, et al. ⁷⁴ show that the PRSKB and PRSice-2 produce very similar distributions (see Figures S13-S15). The PRSKB was able to perform all polygenic risk score calculations using a single command at runtime, whereas PRSice-2 required individual input files for each study. Additionally, the PRSKB is a position-based tool and can handle mislabeled or merged accession numbers. This feature allowed the PRSKB to identify that variant *rs111418223* had been merged with *rs9271192*

and labeled differently between ADNI and Lambert, et al. ³. PRSice-2 was unable to automatically detect that those two variants had been merged because PRSice-2 depends on variant accession numbers. The PRSKB first searches for accession numbers and then looks for chromosome and position pairs to identify associated variants in the target sequence.

Figure 5: ADNI Polygenic Risk Scores using Lambert et al., 2013 GWA Summary Statistics. PRSice-2 (dark grey), and the PRSKB (light grey) scores are shown in part A. PRSice-2 reports polygenic risk scores that center on 0, so 1.0 was added to each PRSice-2 score to put it on the same scale as the PRSKB, which centers polygenic risk scores based on odds ratios around 1.0. The PRSice-2 median score after transformation is 1.05207 and the PRSKB median score is 1.05338. Since a polygenic risk score is a relative score compared to the sample population, we transformed the PRSKB scores (part B) by subtracting 0.00131 to overlap the shape of the distributions when both algorithms report the same median. Since the scores are normally distributed, a Welch's two-sample t-test was used to determine the similarity between the two distributions, which were nearly identical ($t=0.004782$; $P=0.9962$).

2. **Have the authors checked the computing speed of PRSKB calculator, compared to other existing methods/software? For example, if a user wants to upload 500,000 samples genotyped for 40M SNPs, how long would it take to get PRS? The authors should make this clearer in their manuscript (in a table or figure) so that readers/users should be well informed.**

Since the PRSKB allows users to calculate thousands of polygenic risk scores concurrently using data available on our servers, direct comparisons in computational speed/ resource requirements with other existing methods would not yield a fair comparison. Other software (e.g., PRSice-2, LDpred, Lassosum) require users to identify specific GWAS summary statistics and format those summary statistics so that they can use the algorithm. Preprocessing thousands of GWAS summary statistics was not a use case that they were built to handle. The PRSKB reduces that time to manually identify specific studies and gather formatted data to compute polygenic risk scores. We clarified this benefit in the manuscript on line 444:

" The PRSKB was able to perform all polygenic risk score calculations using a single command at runtime, whereas PRSice-2 required individual input files for each study. Additionally, the PRSKB is a position-based tool and can handle mislabeled or merged accession numbers. This feature allowed the PRSKB to identify that variant *rs111418223* had been merged with *rs9271192* and labeled differently between ADNI and Lambert, et al.

³. PRSice-2 was unable to automatically detect that those two variants had been merged because PRSice-2 depends on variant accession numbers. The PRSKB first searches for accession numbers and then looks for chromosome and position pairs to identify associated variants in the target sequence."

3. Does it provide reliability of each PRS? It may be possible that PRSKB will be used to predict the genetic risk of an individual (clinical practice). In this case, the reliability of PRS should be provided.

On line 81, we state "Choosing an appropriate GWA study to calculate polygenic risk scores is paramount to the fidelity of the calculations because the accuracy and predictive power of a polygenic risk score is dependent on the power and scope of the corresponding GWA study data^{29,30}."

Since the PRSKB has similar performance to PRSice-2, it will have similar reliability as previous polygenic risk score calculations. We added a section on line 434 describing the similarity in risk scores calculated using PRSKB and PRSice-2 (see response to issue #1):

We also performed a case study on Alzheimer's disease and show that the polygenic risk scores can separate Alzheimer's disease cases from controls (line 400):

ADNI Case Study

Although we used the GWA summary statistics from Jansen, et al. ² to compare only two groups in the ADNI dataset due to limited sample size for the mild cognitive impairment group (i.e., we combined Alzheimer's disease or mild cognitive impairment versus controls and combined controls or mild cognitive impairment versus Alzheimer's disease), we used an adjusted significance level of 0.01 to account for multiple testing of five potential comparisons of Alzheimer's disease risk: Alzheimer's disease versus mild cognitive impairment; Alzheimer's disease versus controls; mild cognitive impairment versus controls; Alzheimer's disease or mild cognitive impairment versus controls; and mild cognitive impairment or controls versus Alzheimer's disease. A Mann-Whitney U test revealed a significant difference between Alzheimer's disease polygenic risk scores in individuals with a $CDR \geq 1$ and individuals with a $CDR \leq 0.5$ ($P=2.75 \times 10^{-9}$). Similarly, a Mann-Whitney U test also detected a significant difference between Alzheimer's disease polygenic risk scores for individuals with a $CDR=0$ and individuals with any amount of dementia ($CDR \geq 0.5$), although it was less significant ($P=1.97 \times 10^{-7}$). Figures 3 and 4 show the comparisons of polygenic risk score distributions in each CDR cohort. Similar comparisons were made using GWA summary statistics from Lambert, et al. ³ and Lo, et al. ⁷⁴, and are shown in Figures S11 and S12, respectively.

Figure 1: Alzheimer's disease polygenic risk score distribution for ADNI participants with a CDR ≥ 1 and a CDR ≤ 0.5 .

Figure 4: Alzheimer's disease polygenic risk score distribution for ADNI participants with a CDR ≥ 0.5 and ADNI participants with a CDR=0.

After calculating polygenic risk scores from all other studies in the PRSKB database for the individuals in the ADNI cohort and correcting for multiple testing, we identified 42 GWA studies that produced risk scores that significantly differ ($P < 4.21 \times 10^{-6}$) between individuals with and without Alzheimer's disease (see Table S8) and found 29 GWA studies that produced risk scores that significantly differed ($P < 4.23 \times 10^{-6}$) between individuals with cognitive impairment and normal cognition (see Table S9).

- 4. Although a large number of GWAS data are available in PRSKB, they are already in public domain. So, the merit of PRSKB is to collect the large amount of information in one place, which make it easier for users to find reference data (GWAS summary stats). This can be much more useful if users can search relevant data according to their specific purpose, e.g. genotype-by-environment summary stats (please see GIANT consortium homepage). Please also note that a recent paper has introduced a method utilizing genotype-by-environment summary stats. (GxEsum: a novel approach to estimate the phenotypic variance explained by genome-wide GxE interaction based on GWAS summary statistics for biobank-scale data (2021) Genome Biology 22: 183.)**

We provide additional filtering criteria via the interactive web interface to allow users to filter GWAS summary statistics based on disease/trait and study IDs. We link each study directly to the GWAS Catalog, which allows users to view more detailed explanations of each study. The main advantage of the PRSKB is that users can now utilize the latest GWAS summary statistics to calculate the genetic risk for all diseases/traits included in the GWAS Catalog both online and via a command line interface without manually downloading and formatting those files for use in other software packages. We envision that most users will want to perform bulk polygenic risk score analyses using all studies in the database to see if the user-supplied cohorts suffer from an unexpected nonrandom predisposition to any disease or trait. We articulate this vision on line 136: " Because the PRSKB simplifies polygenic risk score calculations and contextualization across thousands of studies that can all be performed with a single command at runtime, we anticipate that this tool will enable a wider adaptation of polygenic risk score calculations through clinical trial screenings, analyses of comorbidities, identifying confounding genetic factors, and various other analyses related to disease genetics."

We also added a citation to GxEsum on line 97: "Polygenic risk scores can also test for gene-by-environment and gene-by-gene interactions^{46,47} through Mendelian randomization studies, which detect causal genetic relationships^{48,49}, and genotype-by-environment interactions based on GWA summary statistics are increasingly common on biobank-scale data⁵⁰."

- 5. There are a number of interfaces that are similar to PRSKB, e.g. Impute.me, Polygenic Score Catalog and so on. The authors should make it clearer what is novel and unique property of PRSKB, compared to those existing resources.**

We added citations to these other efforts on line 117:

" Other notable efforts to centralize polygenic risk scores for research, such as the Polygenic Score Catalog (PGS Catalog)⁵⁶ and Impute.me⁵⁷, have greatly improved the interpretability and dissemination of polygenic risk scores on precomputed data. However, they currently lack the capability of performing high-throughput analyses on user-specific data across all

available studies. Additionally, users are required to select specific studies or traits to analyze *a priori*, which makes data exploration much more time consuming."

The PRSKB is the only interface to systematically provide users with percentile ranks of polygenic risk scores across thousands of polygenic risk score calculations that are computed from a single command at runtime. The PRSKB significantly simplifies the process of calculating risk scores, and that utility has been made clearer throughout the manuscript.

Responses to Reviewer #2:

Reviewer Overview

The authors have developed an online service called PRSKB for calculating polygenic scores using a wide range of GWAS and uploaded target sample data, requiring just a few optional parameters for the user to select. PRSKB uses the p-value thresholding and clumping approach when calculating the polygenic scores, and scales the target sample PRS according to the distribution of PRS found in a range of reference samples. To demonstrate the utility of PRSKB, the authors use PRSKB to calculate Alzheimer's disease PRS in the ADNI cohort, and then test for genetic associations with Alzheimer's disease onset and mild cognitive decline.

I commend the authors on PRSKB. There are many aspects which I think are great, and I think a resource like PRSKB will be useful for researchers. However, I have a number of concerns that should be addressed.

- 1. The LD-clumping process seems unconventional and requires further clarification and perhaps alterations. LD-clumping is typically performed only considering the variants intersecting the GWAS and target data, identifying a list of independent lead variants in the GWAS that are also available in the target sample. In contrast, PRSKB identifies broadly independent 'LD regions' across all variants within a given reference sample, selects one variant within each LD region based on the minimum p-value in the GWAS, and then identifies the variants overlapping with the target sample. I expect this approach will generate suboptimal polygenic scores for two reasons. Retaining only one variant within each LD region may lead to independent genetic effects being excluded due to the lead variants being correlated with other SNPs in common, but uncorrelated with one another. Have the authors considered performing clumping separately for each GWAS?**

LD-clumping is now performed at the sample level and is done for each GWAS separately. We clarified this approach in the manuscript on line 244: "Linkage disequilibrium is then calculated by comparing each locus to the population-specific clumping regions for each GWA study that are housed on our server. "

Comparisons with PRSice-2, which is often used to calculate polygenic risk scores, shows that the PRSKB reports similar results (described on line 433 in the manuscript).

- 2. By performing clumping and selecting lead variants without considering the variants available in the target sample (as indicated by line 211, and figure 2), the variants considered for polygenic scoring may not be present in the target sample, leading to a reduced performance of the polygenic scores. I appreciate that performing clumping specific to the intersect with the target sample is not as practical, and possibly considering the same SNPs for all target sample is advantageous, as the polygenic scores are more comparable across target samples. However, have the authors considered reducing the likelihood of missing data by restricting their polygenic scores to variants that are common and typically available after imputation, such as HapMap3 variants?**

By default, variant selection is performed on the sample to facilitate comparisons between individuals. Minor allele frequencies can be used to impute missing variants based on the sample or available datasets (e.g., 1000 Genomes or UK Biobank). We clarified the clumping in the manuscript.

Line 235:

" The PRSKB first ensures that the summary data and the query data are in the same format (e.g., strand flipping and same reference genome). Next, missing genotypes are imputed based on the minor allele frequency of either the sample or specified dataset (e.g., 1000 Genomes population or UK Biobank) and that frequency is used in the polygenic risk score calculation (e.g., if the minor allele frequency for a missing genotype were 0.2, then the reported risk attributed to that missing genotype would be 0.2 times 2 alleles times the associated risk from the GWA study). An optional parameter allows users to set an imputation threshold that removes studies from the output file where the number of imputed genotypes exceeds a specified percentage. By default, at least half of the genotypes used to calculate the polygenic risk score must be included in the sample. Linkage disequilibrium is then calculated by comparing each locus to the population-specific clumping regions for each GWA study that are housed on our server. When a sample has two or more variants within the same clumping region, the PRSKB chooses the variant with the most significant GWA p-value from that region to represent the clump in the polygenic risk score. The remaining set of independent variants is used in the polygenic risk score calculation."

We revised Figure 2, as follows:

Figure 2: Polygenic risk score workflow. The process follows the standards established by Choi, et al.²²

In addition, we added the following supplementary figure to better illustrate the LD clumping process:

Figure S2: Linkage disequilibrium clumping

Figure S3: Linkage disequilibrium clumping. The LD clumping procedure used by the PRSKB consists of two main parts: 1. Preparation and 2. Processing.

The preparation was performed once, in conjunction with the building of the PRSKB tool. First, we separated the 1000 Genomes samples by population. Then, for each population, we ran PLINK² LD Clumping on each of the samples' variants. Generally, PLINK LD Clumping is used to select a single variant with the most significant association with a given trait from each LD region. However, because our intent was to create general LD regions without regard to any specific trait, we assigned each 1000 Genomes variant a p-value of 0 in order to produce groups of unordered variants. Based on previous polygenic risk score analyses that use the clumping method, we used an r-squared threshold of 0.25 and a kb threshold of 500^{3,4}. After applying this process for both the hg19 and hg38 available reference genome data, we converted the variant coordinates in each LD clump to the hg17 and hg18 reference genomes. Next, we assigned the same LD clump ID number to variants within the same LD region and uploaded this information to the PRSKB database. Linkage disequilibrium (LD) clumping files are stored on our server and are available online by using the population and reference genome. The URL for the European population (EUR) using reference genome hg38 would be written as follows: https://prs.byu.edu/get_clumps_download_file?refGen=hg38&superPop=EUR.

The processing step occurs each time a PRS calculation is performed. First, the PRSKB calculator retains variants that overlap between the base data (the GWAS summary statistics) and query data. Then, for each overlapping variant, the calculator queries the database to retrieve the corresponding LD clump ID based on the population of the base data's samples and the user-specified reference genome. Then, for each sample in the query data, the calculator retains the

variant from each LD region that has the lowest p-value in the base data. This final set of retained variants is then used to calculate a risk score for that sample.

- 3. Line 316: “The user must also indicate the population of the samples to perform accurate linkage disequilibrium clumping”. It appears the LD clumping is based on the ancestry of the target sample. However, it is more appropriate to select the LD clumping reference based on the ancestry of the GWAS sample, as the LD reference aims to recapitulate the LD structure within the GWAS sample to avoid double counting non-independent genetic effects. Can the authors clarify if this is not the case, explain their reasoning if it is the case, or change their methodology to use an LD reference that matches the ancestry of the GWAS sample?**

We altered our methodology to use an LD reference that matches the ancestry of the GWA study sample. We removed the sentence on Line 316 from the manuscript.

- 4. Given my previous comment regarding the increased likelihood of missing variants in the target sample, please can the authors describe how PRSKB handles missing variants when calculating the polygenic scores? A commonly used approach is to use the reference sample allele frequency to impute missing variants. However, to ensure the PRS are of good quality, there should also be an initial check that most variants considered are available in the target sample. Are any such checks provided?**

PRSKB does provide various quality control checks. Missing SNPs for each sample are imputed based on the MAF of specified the population (e.g., 1000 Genomes AFR or UK Biobank), and the user can specify which percentage of SNPs can be imputed (default 50%). The PRSKB output file gives users options to view which SNPs were used in the polygenic risk score calculation. All of these options are described on the PRSKB GitHub README (<https://github.com/kauwelab/PolyRiskScore>) and on readthedocs.io (<https://polyriskscore.readthedocs.io/en/latest/>).

- 5. It seems as though the genetic data of the target sample must be somehow uploaded to the PRSKB server. However, this is often not possible due to data privacy concerns. Or is it possible to download the reference data and carry out of the analysis locally, avoiding sending the data to the PRSKB server? Please can the authors clarify how data privacy concerns are considered, or acknowledge that this is a limitation of their service?**

User data is never uploaded to the PRSKB server. We address this concern on line 181: " Users have two platforms from which they can calculate polygenic risk scores. The first platform is a web interface accessible at <https://prs.byu.edu> via a web browser that allows users to perform client-side calculations where user data are never uploaded to the PRSKB server. The second platform is a command-line interface (CLI) tool that can be run from the Linux or Mac command-line or from a bash shell on Windows."

6. The authors demonstrate how the PRSKB server can be used to generate polygenic scores for research using the ADNI sample. The authors report significant associations and argue that this validates their approach. Whilst this is true to some extent, I think it is important to compare the variance explained by polygenic scores derived using PRSKB with those derived using more conventional approaches, such as PRSice. If the variance explained is similar this would be reassuring that the PRSKB methodology is producing polygenic scores of comparable quality to those based on current practice.

We added a comparison of PRSKB to PRSice-2 and report those results on line 434:

Comparison to PRSice-2

The PRSKB reports similar polygenic risk score results as PRSice-2. Figure 5A plots the polygenic risk scores calculated for both the PRSKB and PRSice-2 across ADNI participants using the Lambert, et al. ³ GWA study. Since polygenic risk scores are a relative measurement of genetic risk compared to a population, we compared the shape of the distributions from the PRSKB and PRSice-2 to ensure that both algorithms report similar score distributions. After performing a minor transformation to have the same median values for both algorithms (original difference between medians is 0.001306), a Welch's two sample t-test shows that slight variations between the two algorithms do not change the overall shape of the distributions (see Figure 5B; $t=0.004782$; $P=0.9962$). Similar comparisons between Alzheimer's disease and cognitive normal controls in the ADNI dataset using GWA studies from Lambert, et al. ³, Jansen, et al. ², and Lo, et al. ⁷⁴ show that the PRSKB and PRSice-2 produce very similar distributions (see Figures S13-S15). The PRSKB was able to perform all polygenic risk score calculations using a single command at runtime, whereas PRSice-2 required individual input files for each study. Additionally, the PRSKB is a position-based tool and can handle mislabeled or merged accession numbers. This feature allowed the PRSKB to identify that variant *rs111418223* had been merged with *rs9271192* and labeled differently between ADNI and Lambert, et al. ³. PRSice-2 was unable to automatically detect that those two variants had been merged because PRSice-2 depends on variant accession numbers. The PRSKB first searches for accession numbers and then looks for chromosome and position pairs to identify associated variants in the target sequence.

Figure 5: ADNI Polygenic Risk Scores using Lambert et al., 2013 GWA Summary Statistics. PRSice-2 (dark grey), and the PRSKB (light grey) scores are shown in part A. PRSice-2 reports polygenic risk scores that center on 0, so 1.0 was added to each PRSice-2 score to put it on the same scale as the PRSKB, which centers polygenic risk scores based on odds ratios

around 1.0. The PRSice-2 median score after transformation is 1.05207 and the PRSKB median score is 1.05338. Since a polygenic risk score is a relative score compared to the sample population, we transformed the PRSKB scores (part B) by subtracting 0.00131 to overlap the shape of the distributions when both algorithms report the same median. Since the scores are normally distributed, a Welch's two-sample t-test was used to determine the similarity between the two distributions, which were nearly identical ($t=0.004782$; $P=0.9962$).

7. Line 76: “polygenic risk scores can capture missing heritability”. Can the authors please elaborate on this statement?

We clarified this statement, as follows (line 82):

"When used appropriately, polygenic risk scores can capture genetic predisposition for diseases or traits across various genetic markers and can be used to assess the genetic risk compared to a specific population³¹⁻³⁴."

8. Line 85: “nominally significant differences in genetic risk scores for bipolar disorder between patients with and without psychotic symptoms suggest a more valid subclassification of the disease”. More valid than what? For a nominally significant association, I thought this statement sounded quite strong.

We removed that sentence.

9. Line 99: “PRSice-2 requires extensive startup time to fully utilize all available options”. Can the authors please explain what they mean by this? Is ‘startup time’ the time taken for someone to understand the software?

We clarified this sentence to read (line 111), "PRSice-2 also has a significant learning curve to be able to understand and utilize the available options, which can limit its application in labs without a strong bioinformatics presence."

10. Line 126: “The data are filtered to include only associations that contain both an odds ratio and the respective risk allele”. Requiring an odds ratio implies PRSKB only include PRS for binary outcomes. Is this the case? If this is a limitation and should clearly highlighted elsewhere.

We have updated this line to show that both odds ratios and beta values are included in the PRSKB (line 149):

"The data are filtered to include only associations that contain both a beta value (or odds ratio) and the respective risk allele."

We updated verbiage throughout the manuscript to reflect this change.

11. Line 128: “any allele that has been reported on the reverse strand is automatically detected and flipped”. Can you clarify whether non-synonymous variants are excluded, and if not, how non-synonymous SNPs are aligned?

Since the polygenic risk score is based on nuclear DNA and is not related to corresponding proteins, non-synonymous variants are treated the same as synonymous variants. In fact, many variants in the dataset are also intronic or intergenic.

12. The website shows that users can upload GWAS summary statistics of their own, but I cannot see where this is described in the paper.

We added the following reference to this functionality in the manuscript (line 232):

"Although a single GWA study is used to calculate each polygenic risk score, users can select multiple studies or traits, which will each be analyzed independently. Users can also use their own GWA summary statistics for personalized analyses."

13. Figure 1. The figure is useful but unfortunately the text is very small. I suggest the authors try to increase the font size.

We increased the font size and revised Figure 1 as follows:

PRSKB Tool Structure

14. Line 208: “When an individual in the target data has two or more variants within the same clumping region, the PRSKB chooses the variant with the most significant p-value”. This sentence makes it sound like variant selection does not consider the variants available in the target sample, in contrast to descriptions in line 211 and Figure 2. Please can the authors clarify. Furthermore, it sounds like variant selection is done at the individual level, rather than the sample level. Presumably this is performed at the sample level, otherwise different variants will be considered for each individual, reducing the comparability of the PRS across individuals. Please can the authors clarify.

By default, variant selection is performed on the sample to facilitate comparisons between individuals. Users may also choose to perform individual-based clumping to utilize the most non-missing genotypes for each individual. However, sample-wide clumping is generally used to facilitate comparisons between individuals. Minor allele frequencies can be used to impute missing variants based on the sample or available datasets (e.g., 1000 Genomes or UK Biobank). We clarified the clumping in the manuscript.

Line 235:

" The PRSKB first ensures that the summary data and the query data are in the same format (e.g., strand flipping and same reference genome). Next, missing genotypes are imputed based on the minor allele frequency of either the sample or specified dataset (e.g., 1000 Genomes population or UK Biobank) and that frequency is used in the polygenic risk score calculation (e.g., if the minor allele frequency for a missing genotype were 0.2, then the reported risk attributed to that missing genotype would be 0.2 times 2 alleles times the associated risk from the GWA study). An optional parameter allows users to set an imputation threshold that removes studies from the output file where the number of imputed genotypes exceeds a specified percentage. By default, at least half of the genotypes used to calculate the polygenic risk score must be included in the sample. Linkage disequilibrium is then calculated by comparing each locus to the population-specific clumping regions for each GWA study that are housed on our server. When a sample has two or more variants within the same clumping region, the PRSKB chooses the variant with the most significant GWA p-value from that region to represent the clump in the polygenic risk score. The remaining set of independent variants is used in the polygenic risk score calculation."

We revised Figure 2, as follows:

Figure 4: Polygenic risk score workflow. The process follows the standards established by Choi, et al.²²

In addition, we added the following supplementary figure to better illustrate the LD clumping process:

Figure S2: Linkage disequilibrium clumping

Figure S5: Linkage disequilibrium clumping. The LD clumping procedure used by the PRSKB consists of two main parts: 1. Preparation and 2. Processing.

The preparation was performed once, in conjunction with the building of the PRSKB tool. First, we separated the 1000 Genomes samples by population. Then, for each population, we ran PLINK² LD Clumping on each of the samples' variants. Generally, PLINK LD Clumping is used to select a single variant with the most significant association with a given trait from each LD region. However, because our intent was to create general LD regions without regard to any specific trait, we assigned each 1000 Genomes variant a p-value of 0 in order to produce groups of unordered variants. Based on previous polygenic risk score analyses that use the clumping method, we used an r-squared threshold of 0.25 and a kb threshold of 500^{3,4}. After applying this process for both the hg19 and hg38 available reference genome data, we converted the variant coordinates in each LD clump to the hg17 and hg18 reference genomes. Next, we assigned the same LD clump ID number to variants within the same LD region and uploaded this information to the PRSKB database. Linkage disequilibrium (LD) clumping files are stored on our server and are available online by using the population and reference genome. The URL for the European population (EUR) using reference genome hg38 would be written as follows: https://prs.byu.edu/get_clumps_download_file?refGen=hq38&superPop=EUR.

The processing step occurs each time a PRS calculation is performed. First, the PRSKB calculator retains variants that overlap between the base data (the GWAS summary statistics) and query data. Then, for each overlapping variant, the calculator queries the database to retrieve the corresponding LD clump ID based on the population of the base data's samples and the user-specified reference genome. Then, for each sample in the query data, the calculator retains the

variant from each LD region that has the lowest p-value in the base data. This final set of retained variants is then used to calculate a risk score for that sample.

- 15. Line 234: “we divided the samples by population region”. I can see how this was done for the 1000 genomes sample since it has reliable ancestry data, but how was this achieved for ADNI and UK Biobank? Self-reported ancestry, or reference-projected principal components?**

We clarified our methodology starting in line 263:

"In order to interpret polygenic risk scores, individual results must be contextualized against a large cohort of similar ethnicity²⁹. The 1000 Genomes Project⁵⁸ contains the best representation of allele frequencies in unrelated individuals across diverse populations and has sequencing data for 2,504 unrelated individuals spanning five superpopulations. We also recognize that some users might want to contextualize their scores against a larger population. Therefore, we also included a separate cohort of 487,409 relatively healthy individuals of primarily European descent from the United Kingdom (UK) Biobank⁵⁹. We used the PRSKB to compute polygenic risk scores from all GWA studies in our database for each individual in each cohort (each 1000 Genomes population was a different cohort). We then calculated the percentile rank of each person against all other people in the cohort. The polygenic risk score and percentile ranks were passed to Plotly JavaScript⁷¹ to create interactive graphics that allow users to visualize population-specific distributions of polygenic risk scores for any study in the PRSKB database. Dynamic plots with a table of summary statistics for each study are available for users to query online at <https://prs.byu.edu/visualize.html>."

- 16. Line 257: “collective significant relationship between age, sex, and polygenic risk score”. I assume this means the regression was $PRS \sim Age + Sex$, but please can the authors clarify.**

We removed this sentence for clarity.

- 17. Line 259: “polygenic risk score outcomes”. What is a polygenic score outcome? Are you simply saying there was no association between polygenic scores and sex or age? Please clarify.**

We removed this sentence for clarity.

- 18. Lines 313-315: “Next, the user must specify the reference genome (hg38, hg19, hg18, or hg17) used to sequence the input variants so that the associations queried from the**

database correspond to the same reference assembly". Presumably this is only necessary when using VCF format, since RSIDs are not specific to genome build? Please clarify.

We clarified that the genome build is required only for VCF file input. Line 341 was changed as follows:

" Next, the user must specify the reference genome (hg38, hg19, hg18, or hg17) used to sequence the input variants if they are using the VCF file format so that the associations queried from the database correspond to the same reference assembly. By default, hg38 is used as a reference for RSIDs."

19. Line 319: "the largest study cohort measured by the initial sample size plus the replication sample size". Can the authors please clarify why it is useful to filter GWAS by the discovery + replication sample size? It seems more appropriate to filter GWAS by the sample size in the GWAS summary statistics available, which often does not include the replication sample.

The cohort size is reported as the sum of the initial sample size plus the replication sample size because the GWAS Catalog often reports only combined p-values for the initial and replication data. While not all studies report a combined p-value, it is impossible to distinguish the type of p-value from the GWAS catalog statistics, so we have chosen our current approach. More information can be found at the following link from under the "Top Associations" heading: <https://www.ebi.ac.uk/gwas/docs/methods/criteria>

20. The authors test for PRS associations with Alzheimer's disease phenotypes in part to demonstrate PRSKB works, but they also use the results for some inference about the etiology of Alzheimer's disease. However, it does not appear the authors have accounted for population stratification when performing these analyses, by using principal components of ancestry as covariates, or linear mixed models. Can the authors highlight this as a limitation or account for ancestry as is standard when performing polygenic score analysis for inference?

We added the following sentences on line 290: " We used all 808 whole-genome sequences from the ADNI cohort that also have a clinical dementia rating (CDR) score (see Table S4 for the number of samples in each CDR group). Population structure was previously analyzed⁷² and shows that the ADNI whole-genome sequencing participants are primarily similar to the European population in the 1000 Genomes Project."

21. Line 450: "First, while the PRSKB contains over 17,900 GWA variants, haplotype associations, or associations that include multiple variants for a single effect size, were removed from the PRSKB database". I think this sentence could be clarified by making a clearer distinction between what is and isn't included PRSKB. For example "...contains over 17,900 GWA variants, we removed haplotype associations..."

We have clarified this point. Line 150: "Each variant is analyzed independently (i.e., risk haplotypes are excluded)."

Line 508: "For example, we remove multi-allele haplotype associations from the PRSKB database and ensure that combinations of multiple variants cannot have a single effect. The PRSKB analyzes each variant individually."

Responses to Reviewer #3:

Reviewer Overview

This manuscript describes the Polygenic Risk Score Knowledge Base (PRSKB), an online database and server for calculating polygenic scores from a large number of GWAS summary statistics originating the GWAS Catalog.

Overall, the manuscript is well written and represents a very large effort in collating and building this resource. This is a valuable contribution to the human genetics research community.

I have several comments/questions to help clarify the analysis and how the tool works.

1. Even though the ADNI Alzheimer's analysis is simply an example and not the main focus of the manuscript, more information is needed:

a. Some information about the ADNI study design (a table would be useful): how many individuals, sex, number of cases/controls, ages, etc.

We added Table S4 to the supplement:

	Alzheimer's Disease (CDR\geq1.0)	Mild Cognitive Impairment (CDR=0.5)	Cognitive Normal (CDR=0.0)
Number of Subjects	592	98	118
Female Count (%)	245 (41.39%)	56 (57.14%)	61 (51.70%)
Average Age	73.28 \pm 7.35	73.03 \pm 7.04	73.17 \pm 5.59

We also added the following sentence with reference to more in-depth analyses of the ADNI cohort (line 276): "Population structure was previously analyzed⁷⁰ and shows that the ADNI whole-genome sequencing participants are primarily similar to the European population in the 1000 Genomes Project."

b. More details about the Alzheimer's PRS are needed: how many SNPs were used, was APOE4 included?

We added the following information to the main manuscript and supplement (line 290):

" We used all 808 whole-genome sequences from the ADNI cohort that also have a clinical dementia rating (CDR) score (see Table S4 for the number of samples in each

CDR group). Population structure was previously analyzed⁷² and shows that the ADNI whole-genome sequencing participants are primarily similar to the European population in the 1000 Genomes Project. CDR is a summary measure developed to denote the overall severity of dementia in an individual, where CDR=0 is considered normal cognition, CDR=0.5 is mild cognitive impairment, and CDR≥1.0 is Alzheimer's disease⁷³. As a case study, we used the PRSKB calculator to compute the polygenic risk scores for each ADNI participant for three Alzheimer's disease GWA studies available in our database: Lambert, et al.³, Jansen, et al.², and Lo, et al.⁷⁴. The genetic variants used for each polygenic risk score calculation are listed in Tables S5-S7. The PRSKB imputed missing genotypes using the entire ADNI cohort minor allele frequency and used variant linkage disequilibrium based on the European population in the 1000 Genomes Project."

- c. Based on Fig 3, the PRS distribution was not substantially deviating from normality but there were some small noticeable deviations, particularly for individuals with CDR≥1. It could be due to the ascertainment bias of ADNI (case/control), population stratification, batch effects, or the strong effect of APOE4 alleles relative to the rest of the SNPs in the score. What does the PCA plot of ADNI look like, and were analyses adjusted for principal components / batch effects (if there were any)?**

See response above. The PCA plots for the 808 whole-genome sequences were previously published (reference 73), and we link to that study. To facilitate cross-study comparisons, we opted to include all 808 individuals.

- 2. It's not clear, is the option to show percentiles for UK Biobank or other datasets available also for user uploaded PRS or only for the precomputed PRS?**

We offer visualization for precomputed percentiles, but users may choose to calculate the percentiles for their data as compared to an available dataset, which is reported in the verbose output file. We have added the following sentence for clarification (line 396): "At this time, visualizations on the website are exclusively for pre-computed scores and user-uploaded data are not graphed. However, percentile data can be found for user-uploaded data in the verbose output file."

- 3. Do the UK Biobank analyses only include the British whites or are all participants included?**

We clarified our methodology starting in line 263:

" In order to interpret polygenic risk scores, individual results must be contextualized against a large cohort of similar ethnicity²⁹. The 1000 Genomes Project⁵⁸ contains the best representation of allele frequencies in unrelated individuals across diverse populations and has sequencing data for 2,504 unrelated individuals spanning five superpopulations. We

also recognize that some users might want to contextualize their scores against a larger population. Therefore, we also included a separate cohort of 487,409 relatively healthy individuals of primarily European descent from the United Kingdom (UK) Biobank⁵⁹. We used the PRSKB to compute polygenic risk scores from all GWA studies in our database for each individual in each cohort (each 1000 Genomes population was a different cohort). We then calculated the percentile rank of each person against all other people in the cohort."

- 4. Is there a filter for minor allele frequency to ensure that only common variants are included in the PRS? Some GWAS include rare variants as well which could end up in the calculation.**

An optional filter allows users to use only variants that exceed a specified minor allele frequency. By default, we include all associated genetic variants identified in GWAS. Recent studies suggest that integrating rare variants in polygenic risk scores improves the accuracy of the polygenic risk score compared with filtering for just common variants. We added a sentence with a reference to rare variants helping polygenic risk score calculations to line 521: "Incorporating rare variants in polygenic risk score calculations actually improves polygenic risk score prediction⁸³, and the PRSKB uses all associated variants in its calculations by default, with an optional parameter to filter variants based on their minor allele frequencies."

- 5. Line 46, 'Over 71,000 genetic loci are currently implicated in diseases or traits' is not quite right, these are pairs of associations, not loci (each locus can be associated with multiple traits).**

The sentence now reads (line 55), "Tens of thousands of genetic associations are currently implicated in diseases or traits with genome-wide significance ($p\text{-value} < 5 \times 10^{-8}$)¹, and additional associations have been discovered through meta-analyses²⁻⁴."

- 6. Line 54 'However, a major limitation of GWA studies is that each genetic locus is evaluated individually. Polygenic risk scores address this issue by aggregating all effects', it's not quite right. Many/most PRS are based on GWAS, it's just that the goal is different: GWAS is focused on discovery of trait associations, PRS is about quantifying the total genetic liability towards a specific trait/disease (as discussed later).**

We altered the section as follows (line 62): "GWA studies are effective at identifying individual genetic locus-trait associations. However, GWA results on their own cannot determine the total genetic liability for a given trait in a genome of interest. Polygenic risk scores utilize GWA summary statistics to quantify the aggregate genetic risk for a disease or trait based on all associated genetic variants present in a genome²¹."

- 7. 'For instance, a study of hypertrophic cardiomyopathy in African Americans misclassified benign genetic variants as pathogenic, which could have been avoided if even a small**

number of African Americans were included in the control group' -> is this from a study of common variants or rare variants? Typically, common variants are not labelled 'pathogenic'; a different example could be useful here since the PRS is mainly based on common variants.

We modified the passage (line 68): "Accordingly, polygenic risk scores are dependent on the underlying summary statistics from a GWA study. However, most large-scale GWA studies have been conducted on predominantly European populations²², with results that often do not translate to other populations²³ due to differences in allele frequencies and linkage disequilibrium patterns²⁴⁻²⁶. For instance, effect sizes reported in GWA studies performed primarily on populations of European descent were found to be significantly higher than corresponding effect sizes reported by GWA studies consisting entirely of non-European individuals²⁷. The lack of diversity in GWA study cohorts can also cause important risk alleles in minority populations to remain unidentified. For example, the Population Architecture using Genomics and Epidemiology (PAGE) study found that a novel risk variant associated with the number of cigarettes smoked per day existed at a frequency of 17.2% in Native Hawaiian participants but was absent or rare in most other populations²⁸."

8. The PGS Catalog (<https://pgscatalog.org>) should be mentioned somewhere, it is highly relevant too.

We added a citation to the PGS Catalog on line 117:

" Other notable efforts to centralize polygenic risk scores for research, such as the Polygenic Score Catalog (PGS Catalog)⁵⁶ and Impute.me⁵⁷, have greatly improved the interpretability and dissemination of polygenic risk scores on precomputed data. However, they currently lack the capability of performing high-throughput analyses on user-specific data across all available studies. Additionally, users are required to select specific studies or traits to analyze *a priori*, which makes data exploration much more time consuming."

9. Line 127 'The data are filtered to include only associations that contain both an odds ratio and the respective risk allele.' -> I assume PRSKB is not limited only to case/control but can do continuous traits too, in which case it wouldn't be an odd ratio.

Correct. We revised this section to address how the PRSKB also utilizes beta values as follows (line 149): "The data are filtered to include only associations that contain both a beta value (or odds ratio) and the respective risk allele."

10. 'Additionally, only non-haplotype associations that reside on an autosomal' -> what does 'haplotype' mean here, HLA alleles?

We clarified this sentence (line 150): "Each variant is analyzed independently (i.e., risk haplotypes are excluded)." We also added line 508: "There are certain limitations to the PRSKB. For example, we remove multi-allele haplotype associations from the PRSKB"

database and ensure that combinations of multiple variants cannot have a single effect. The PRSKB analyzes each variant individually."

11. I suggest replacing 'gender' with 'sex' (e.g., Line 457) since for the most part we are dealing with biological sex effects and not gender per se.

We removed all references to 'gender' and replaced them with 'sex.'

We appreciate the positive and constructive feedback that has significantly improved the quality of our tool and manuscript. Thank you for the thoughtful reviews.

Sincerely,

John S.K. Kauwe, Ph.D.

Brigham Young University

Reviewers' comments:

Reviewer #1 (Remarks to the Author):

The authors have properly addressed my previous concerns and the manuscript has been significantly improved. I have no further comments.

Reviewer #2 (Remarks to the Author):

I thank the authors for addressing my comments. There are still a few issues that have not been fully addressed. Please see my attached comments.

Dear Reviewer 2,

Thank you for your thorough reviews to ensure that our manuscript meets the highest standard for publication. We address the three points that you raised below in highlighted text and highlighted our revisions in the main manuscript.

I thank the authors for addressing my comments. There are still a few issues that have not been fully addressed. I have written my additional comments in red below the relevant part of the authors rebuttal.

- 1. The authors demonstrate how the PRSKB server can be used to generate polygenic scores for research using the ADNI sample. The authors report significant associations and argue that this validates their approach. Whilst this is true to some extent, I think it is important to compare the variance explained by polygenic scores derived using PRSKB with those derived using more conventional approaches, such as PRSice. If the variance explained is similar this would be reassuring that the PRSKB methodology is producing polygenic scores of comparable quality to those based on current practice.**

We added a comparison of PRSKB to PRSice-2 and report those results on line 434:

Comparison to PRSice-2

The PRSKB reports similar polygenic risk score results as PRSice-2. Figure 5A plots the polygenic risk scores calculated for both the PRSKB and PRSice-2 across ADNI participants using the Lambert, et al. ³ GWA study. Since polygenic risk scores are a relative measurement of genetic risk compared to a population, we compared the shape of the distributions from the PRSKB and PRSice-2 to ensure that both algorithms report similar score distributions. After performing a minor transformation to have the same median values for both algorithms (original difference between medians is 0.001306), a Welch's two sample t-test shows that slight variations between the two algorithms do not change the overall shape of the distributions (see Figure 5B; $t=0.004782$; $P=0.9962$). Similar comparisons between Alzheimer's disease and cognitive normal controls in the ADNI dataset using GWA studies from Lambert, et al. ³, Jansen, et al. ², and Lo, et al. ⁷⁴ show that the PRSKB and PRSice-2 produce very similar distributions (see Figures S13-S15). The PRSKB was able to perform all polygenic risk score calculations using a single command at runtime, whereas PRSice-2 required individual input files for each study. Additionally, the PRSKB is a position-based tool and can handle mislabeled or merged accession numbers. This feature allowed the PRSKB to identify that variant *rs111418223* had been merged with *rs9271192* and labeled differently between ADNI and Lambert, et al. ³. PRSice-2 was unable to automatically detect that those two variants had been merged because PRSice-2 depends on variant accession numbers. The PRSKB first searches for accession numbers and then looks for chromosome and position pairs to identify associated variants in the target sequence.

Figure 5: ADNI Polygenic Risk Scores using Lambert et al., 2013 GWA Summary Statistics. PRSice-2 (dark grey), and the PRSKB (light grey) scores are shown in part A. PRSice-2 reports polygenic risk scores that center on 0, so 1.0 was added to each PRSice-2 score to put it on the same scale as the PRSKB, which centers polygenic risk scores based on odds ratios around 1.0. The PRSice-2 median score after transformation is 1.05207 and the PRSKB median score is 1.05338. Since a polygenic risk score is a relative score compared to the sample population, we transformed the PRSKB scores (part B) by subtracting 0.00131 to overlap the shape of the distributions when both algorithms report the same median. Since the scores are normally distributed, a Welch's two-sample t-test was used to determine the similarity between the two distributions, which were nearly identical ($t=0.004782$; $P=0.9962$).

Thank you for including this comparison. However, this does not compare the phenotypic variance explained by polygenic scores from PRSKB and PRSice, which is an important validation step for their tool. Comparing the mean of each PRS using a t-test does not show the variance explained by the score will be equivalent.

We added the following text to the main manuscript (line 450): "Additionally, we found similar phenotypic variance explained by the PRSKB and PRSice-2 in ADNI when using associated variants in each of the three Alzheimer's disease genome-wide association studies (see Table S10)."

Table S10:

Group 1	Group 2	Study Authors	Study ID	Trait	Statistical Test	PRSKB Test Statistic	PRSKB P-value	PRSice-2 Test Statistic	PRSice-2 P-value
CDR=1	CDR≤0.5	Lambert, et al.	GCST002245	Alzheimer's disease	Mann Whitney U Test	68160	0.002295184	39451	0.002863105
CDR=1	CDR≤0.5	Jansen, et al.	GCST007320	Alzheimer's disease	Mann Whitney U Test	71392	2.60584E-05	35914	1.07586E-05
CDR=1	CDR≤0.5	Lo, et al.	GCST009496	Alzheimer's Disease	Mann Whitney U Test	74035	2.59068E-07	36471	1.1125E-07
CDR≥0.5	CDR=0	Lambert, et al.	GCST002245	Alzheimer's disease	Welch's Two-Sample T-Test	1.74402877	0.083138866	1.895648761	0.059849022
CDR≥0.5	CDR=0	Jansen, et al.	GCST007320	Alzheimer's disease	Mann Whitney U Test	45651	0.000338969	27829	0.000252806
CDR≥0.5	CDR=0	Lo, et al.	GCST009496	Alzheimer's disease	Mann Whitney U Test	48483	1.18525E-06	27211.5	6.29065E-07

A Welch's two-sample t-test was used for one comparison because the scores were normally distributed.

- Line 128: "any allele that has been reported on the reverse strand is automatically detected and flipped". Can you clarify whether non-synonymous variants are excluded, and if not, how non-synonymous SNPs are aligned?

Since the polygenic risk score is based on nuclear DNA and is not related to corresponding proteins, non-synonymous variants are treated the same as synonymous variants. In fact, many variants in the dataset are also intronic or intergenic.

Apologies, I meant to say 'ambiguous' instead of 'synonymous'. Please can the authors confirm whether they exclude synonymous variants, i.e. A/T or C/G alleles, where the strand cannot be determined.

We clarified how ambiguous variants are used to calculate polygenic risk scores (starting at line 153): "Finally, any allele that has been reported on the reverse strand is automatically detected and flipped to the forward strand. The strand-flipping procedure entails comparing each reported risk allele to

the list of possible alleles for the specified variant from dbSNP⁶². If the reported risk allele does not exist in the list of possible alleles, the complement of the risk allele is checked against the dbSNP list. If the complement is present, then it is used as the reported risk allele for polygenic risk score calculations, as recommended by Choi, et al.²¹. Ambiguous variants that cannot be resolved by this method are used as originally reported in the genome-wide association study."

- 3. The authors test for PRS associations with Alzheimer's disease phenotypes in part to demonstrate PRSKB works, but they also use the results for some inference about the etiology of Alzheimer's disease. However, it does not appear the authors have accounted for population stratification when performing these analyses, by using principal components of ancestry as covariates, or linear mixed models. Can the authors highlight this as a limitation or account for ancestry as is standard when performing polygenic score analysis for inference?**

We added the following sentences on line 290: " We used all 808 whole-genome sequences from the ADNI cohort that also have a clinical dementia rating (CDR) score (see Table S4 for the number of samples in each CDR group). Population structure was previously analyzed⁷² and shows that the ADNI whole-genome sequencing participants are primarily similar to the European population in the 1000 Genomes Project."

If the authors will not control for population structure, they must provide more specific evidence to show it is not necessary. It is best practice to include principal components of population structure as covariates even when all individuals are of European ancestry. I have looked back at the reference provided by the authors (Osipowicz et al.) and could not see any evidence to suggest controlling for population structure was not necessary. I am aware controlling for population structure in this context is unlikely to substantially change the results but I think the authors should provide a better justification.

The citation refers to the principal component analysis in Supplementary Figure S1 (https://oup.silverchair-cdn.com/oup/backfile/Content_public/Journal/nargab/3/3/10.1093_nargab_lqab069/1/lqab069_supplemental_file.pdf?Expires=1656680471&Signature=rdaKQgEBxh~R2pHVh4s41iovhxVUhcCQCGa9kLd7E7kO3TZS3oYWiuUjVDcA7WTZM1MvJORbJeC0isLE9MoXdOx26AXevOj195kJRoVE-SrZb~R30fJm9IhZbzEewldjcS4vF~NZWdRFB9WWefgIKLP-8R3CD7hY90quG45AAIKmIWtOxxNvKAie3vh1smDlusBKKnLe4JLrpBW87r7Gebm3gWXPevV-Y7zPWmvLrdHKzwezKd-V9jCasfFdE0-9REepISSQHfC-ZoryQC07wuTIsI3hW7V0kvLp59rl2f8g2zlrGnwAnAv1757x6sMTOUtp4nmqbg4jA0iZEoKq6A&Key-Pair-Id=APKAIE5G5CRDK6RD3PGA), which we say "shows that the ADNI whole-genome sequencing participants are primarily similar to the European population in the 1000 Genomes Project."

We added the following explanation of population structure on line 290:

We recognize that uncorrected population structure can either inflate or deflate polygenic risk score associations when the population structure of the base and target samples significantly differ²¹. Inaccurate adjustments for population structure can also introduce biases into polygenic risk scores²¹. We decided not to correct for population structure in ADNI because (1) the population structure for the base data from the genome-wide

association studies included in the GWAS Catalog indicate general geographic locations for the included subjects without including principal components, and (2) the principal component analysis of the ADNI whole genome sequences shows that the population structure of ADNI is largely similar to the general geographic location of the base data. Both the PRSKB and PRSice-2 were run using the same assumptions to ensure that the results are directly comparable.

REVIEWERS' COMMENTS:

Reviewer #2 (Remarks to the Author):

Thank you for addressing my comments. I have no further comments.